# Conservation of regulatory elements with highly diverged sequences across large evolutionary distances

**Mai H. Q. Phan**[1,2,8], **Tobias Zehnder** [2,8], **Fiona Puntieri**[2], **Andreas Magg** [1,2], **Blanka Majchrzycka**[1,2], **Milan Antonović**[1,2,3], **Hannah Wieler**[1,2,3], **Bai-Wei Lo**[2], **Damir Baranasic** [4,5,6], **Boris Lenhard** [5,6], **Ferenc Müller** [7], **Martin Vingron** [2] & **Daniel M. Ibrahim** [1,2] ✉

Developmental gene expression is a remarkably conserved process, yet most *cis*-regulatory elements (CREs) lack sequence conservation, especially at larger evolutionary distances. Some evidence suggests that CREs at the same genomic position remain functionally conserved independent of sequence conservation. However, the extent of such positional conservation remains unclear. Here, we profiled the regulatory genome in mouse and chicken embryonic hearts at equivalent developmental stages and found that most CREs lack sequence conservation. To identify positionally conserved CREs, we introduced the synteny-based algorithm interspecies point projection, which identifies up to fivefold more orthologs than alignment-based approaches. We termed positionally conserved orthologs 'indirectly conserved' and showed that they exhibited chromatin signatures and sequence composition similar to sequence-conserved CREs but greater shuffling of transcription factor binding sites between orthologs. Finally, we validated indirectly conserved chicken enhancers using in vivo reporter assays in mouse. By overcoming alignment-based limitations, we revealed widespread functional conservation of sequence-divergent CREs.

Embryonic development is driven by deeply conserved sets of transcription factors (TFs) and signaling molecules that control tissue patterning, cell fates and morphogenesis. During the phylotypic stage and organogenesis, tissue-specific and lineage-specific gene expression patterns are similar, even between distantly related organisms[1,2]. For example, in the developing heart, patterning and morphological changes are conserved across vertebrates. The same key TFs in cardiac mesoderm are required in the two-chambered hearts of fish and the four-chambered hearts of birds and mammals[3], arguing for a common gene regulatory basis of embryonic development.

However, most *cis*-regulatory elements (CREs) detected through DNA accessibility or chromatin modifications are not sequence conserved[4,5], especially at larger evolutionary distances. For example, enhancer sequences identified by chromatin marks in embryonic heart are poorly conserved[6]. Similar observations have been made for transcription factor-binding sites (TFBS) in vertebrate livers[7]. Yet, there

[1]Berlin Institute of Health at Charité – Universitätsmedizin Berlin, Center for Regenerative Therapies, Berlin, Germany. [2]Max Planck Institute for Molecular Genetics, Berlin, Germany. [3]Institute of Chemistry and Biochemistry, Freie Universität Berlin, Berlin, Germany. [4]Division of Electronics, Ruder Boskovic Institute, Zagreb, Croatia. [5]MRC Laboratoy of Medical Sciences, London, UK. [6]Institute of Clinical Sciences, Faculty of Medicine, Imperial College London, Hammersmith Hospital Campus, London, UK. [7]Department of Cancer and Genomic Sciences, Birmingham Centre for Genome Biology, School of Medical Sciences, College of Medicine and Health, University of Birmingham, Birmingham, UK. [8]These authors contributed equally: Mai H. Q. Phan, Tobias Zehnder. ✉e-mail: daniel.ibrahim@bih-charite.de

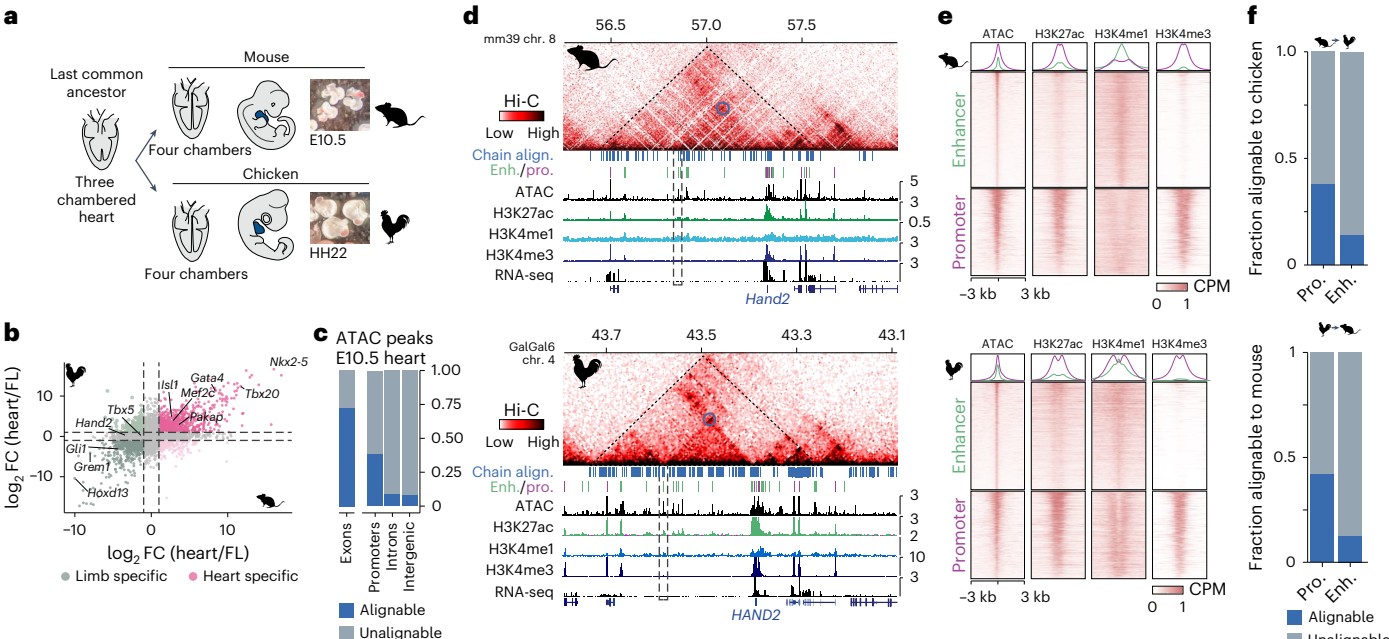

**Fig. 1 | Evolutionary conservation of gene expression and chromatin structure between mouse and chicken embryonic hearts despite divergent CREs. a**, Reptilian and mammalian lineages convergently evolved fully separated four-chambered hearts. E10.5 and HH22 represent equivalent stages of heart formation. **b**, Conservation of global gene expression (log$_2$-transformed fold change (FC) of heart-expressed versus limb-expressed genes) between mouse (E10.5) and chicken (HH22). **c**, ATAC-seq peaks (E10.5 heart) were mostly alignable (LiftOver (minMatch = 0.1)) to chicken in coding but not noncoding regions. **d**, Syntenic regions at the *Hand2/HAND2* locus shows conserved 3D chromatin structure and histone modifications relative to the target gene despite different genomic size. Coverage track unit, counts per million. Dashed triangle indicates conserved topological domain structure, blue circles and dashed rectangle show specific contacts to conserved enhancers. Blue ticks indicate conserved sequences and green or purple ticks indicate predicted promoters or enhancers. **e**, Signal enrichment (± 3kb) of histone modifications at heart promoters (pro.) and enhancers (enh.) (E10.5, HH22), centered on ATAC-seq summits. **f**, Fraction of alignable elements identified in **e** with the chicken or mouse genome (LiftOver (minMatch = 0.1)). Align., alignment; CPM, counts per million.

are several examples of functionally conserved CREs in the absence of sequence conservation[8–10]. For example, the well-known *even-skipped* stripe 2 enhancer shows functional conservation among insects despite highly divergent sequences[11–13].

Determining orthologous CREs in distantly related species is complicated for several reasons. First, rapid turnover of noncoding sequences limits the effectiveness of pairwise alignments. Second, alignment-free methods struggle to accurately determine ortholog pairs. Some alignment-free methods search for similar clusters of TFBS or 'sequence words' as footprints of regulatory elements[14–17], whereas others use machine learning algorithms to successfully identify cell-type-specific enhancers across species. Although this highlights conservation of regulatory sequence information[18–21], ortholog pairing requires separate processing steps. Third, the computational demands and availability of genome assemblies limit the use of multiple-genome alignments, which is an alternative better suited to the task of orthology tracing across species. For example, the zebrafish ortholog of a human limb enhancer was identified indirectly through iterative pairwise alignment between human and spotted gar, and between spotted gar and zebrafish[22]. More systematically, the use of one bridging species (*Xenopus*) helped to uncover hundreds of 'covert' ortholog pairs between human and zebrafish[23]. Approaches using Cactus multispecies alignments of hundreds of genomes[24–27] aim to trace orthology from genome sequences alone. However, these approaches require computational infrastructure and availability of genome assemblies, and they currently cannot bridge larger evolutionary distances (for example, chicken–mouse).

Here, we present an experimental–analytical framework to identify orthologous CREs by combining two currently underutilized features: synteny and functional genomic data. In genomics, synteny describes the maintenance of colinear genomic sequences on chromosomes of different species[28,29]. Not only genes are maintained in synteny; developmental genes are often flanked by conserved noncoding

elements (CNEs), many of which act as enhancers[30–32]. Their syntenic arrangement reflects conserved regulatory environments termed genomic regulatory blocks (GRBs)[29,33]. Functional genomic data, such as chromatin accessibility and histone modifications, are widely used to detect putative CREs. Given that the hearts of birds and mammals are evolutionarily homologous structures, their regulatory genomes should be related. Therefore, experimentally identified CREs from both species might provide the genomic footprint of functionally conserved orthologs whose sequences have diverged to the point at which alignment fails. Here, we use chromatin profiling from murine and chicken hearts at equivalent developmental stages to determine regulatory elements. We then apply interspecies point projection (IPP), a synteny-based algorithm designed to map corresponding genomic locations in highly diverged genomes. Using this strategy, we uncover thousands of previously hidden conserved CREs based on their relative position in the genome. We term these sequence-diverged orthologs 'indirectly conserved' (IC) and compare their functional conservation with that of classical sequence-conserved elements. We find similar enrichment of chromatin marks and, using machine learning (ML) models and TFBS analysis, show that both classes display similar heart-enhancer-specific sequence composition. Finally, we demonstrate functional orthology using in vivo enhancer–reporter assays. However, IC orthologs show higher rearrangement of shared TFBS, preventing detection through sequence alignment. Together, the results of this study demonstrate currently underrepresented widespread conservation of CREs with highly diverged sequences across large evolutionary distances.

## Results

### Identification of embryonic heart CREs in mouse and chicken

To identify CREs driving gene expression at equivalent stages of heart development[34,35], we generated comprehensive chromatin and gene

expression profiles from embryonic mouse and chicken hearts at embryonic days (E) 10.5 and E11.5 and Hamburger Hamilton stages (HH) 22 and HH24 using chromatin immunoprecipitation with sequencing library preparation by Tn5 transposase (ChIPmentation), assay for transposase-accessible chromatin using sequencing (ATAC-seq), RNA sequencing (RNA-seq) and high-throughput chromatin conformation capture (Hi-C) (Fig. 1a). We first compared differentially expressed genes in the heart versus limb in mouse and chicken at E10.5 and HH22 (Fig. 1b). Consistent with previous reports[3], tissue-specific expression was conserved, including that of key TF genes specific for heart and limb development (Fig. 1b and Extended Data Fig. 1a). To characterize conservation of regulatory regions driving this expression, we estimated sequence conservation using LiftOver[36] of mouse E10.5 ATAC-seq peaks in chicken. Most noncoding peaks lacked sequence conservation, in contrast to those overlapping with exons (Fig. 1c and Extended Data Fig. 1b). We then used Hi-C and ChIPmentation to comprehensively profile the regulatory genome. Hi-C confirmed conservation of the 3D chromatin structures overlapping developmentally associated GRBs (Fig. 1d and Extended Data Fig. 1c), as well as enrichment of synteny breaks at topologically associating domain (TAD) boundaries (Extended Data Fig. 1d). Syntenic regions surrounding developmental genes showed comparable distribution of chromatin marks, indicating that the position of regulatory elements relative to their targets might be conserved (Fig. 1d). To establish a high-confidence set of heart enhancers and promoters for both species, we used CRUP to predict CREs from histone modifications for each species[37]. To minimize false positives, we integrated CRUP predictions with chromatin accessibility and gene expression data (Methods). As CREs from E10.5 and E11.5 and those from HH22 and HH24 largely overlapped (Extended Data Fig. 1e), we used the union set of CREs for each species for further analyses. In total, we called 20,252 promoters and 29,498 enhancers in mouse and 14,806 and 21,641 in chicken hearts.

We then estimated sequence conservation for this high-confidence set of CREs. Consistent with previous reports[6], fewer than 50% of promoters and only ~10% of enhancers were sequence conserved (LiftOver (minMatch = 0.1)) between mouse and chicken (Fig. 1f and Extended Data Fig. 1f). Thus, the lack of sequence alignability remained consistent even when the analysis was restricted to a stringently filtered set of enhancers and promoters.

## A synteny-based algorithm identifies ortholog genomic regions

DNA sequence conservation alone is likely to underestimate conserved regulatory regions. To identify such conserved, nonalignable CREs, we developed IPP[38], a synteny-based algorithm designed to identify orthologous positions in two genomes independent of sequence divergence (Fig. 2a and Extended Data Fig. 2). We assumed that any nonalignable element in one genome located between flanking blocks of alignable regions would be located at the same relative position in another genome. Thus, for a given species pair we can interpolate the position of an element (for example, an enhancer) relative to adjacent alignable regions, so-called anchor points. We refer to the interpolated coordinates in the target genome as projections. As a larger distance to an anchor point reduces the accuracy of the projection, the second pillar of IPP involves the use of bridged alignments[22,23]. IPP uses not one but multiple bridging species, increasing the number of anchor points, which minimizes the distance to an anchor point (Fig. 2b). Therefore, we can use IPP to classify projections by their distance to a bridged alignment or direct alignment. We defined parameters to distinguish high-, medium- and low-confidence projections. Regions projected within 300 bp of a direct alignment were defined as directly conserved (DC). Those that were further than 300 bp from a direct alignment but could be projected through bridged alignments were defined as IC regions, if the summed distance to anchor points was less than 2.5 kb. The remaining low-confidence projections were defined

as nonconserved (NC) (Extended Data Fig. 2; for parameterization, see the Supplementary Note).

## IPP improves ortholog detection in distantly related species

To optimize mouse–chicken projections, we selected 16 species, comprising mouse, chicken and 14 bridging species from reptilian and mammalian lineages with ancestral vertebrate or chordate genomes as outgroups. After building our collection of anchor points from pairwise alignments, we projected mouse heart CREs to chicken and all bridging species to estimate their positional conservation at varying evolutionary distances.

The proportion of mouse CREs classified as DC reduced drastically with increasing evolutionary distance. In the closely related rat, more than 90% of CREs were classified as DC, but this proportion dropped to 50–70% within placental mammals and was even lower in nonmammalian vertebrates. In chicken, only 22% of promoters and 10% of enhancers were identified as DC (Fig. 2c). IPP identified additional orthologs through IC regions. Within distantly related vertebrates in particular, this substantially increased putatively conserved CREs (orange fraction, Fig. 2c). For the mouse–chicken comparison, positionally conserved promoters increased more than threefold (from 18.9% (DC) to 65% (DC + IC)) and enhancers more than fivefold (7.4% to 42%). With these increases, IPP paired an additional 8,138 promoters and 9,699 enhancers with candidate orthologs in chicken.

We compared IPP with LiftOver as a reference for sequence conservation. In practice, LiftOver performed similarly in terms of numbers to IPP DC projections (Fig. 2c). For the mouse–chicken comparison, the vast majority (>84%) of LiftOver-identified orthologs were also identified by IPP as DC or IC (Extended Data Fig. 3a,b). However, only 53% of promoters and 24% of enhancers identified by IPP corresponded to orthologs detected by LiftOver. Although similar in numbers, DC regions were not interchangeable with LiftOver-identified orthologs. LiftOver considers the entire CRE sequence (here 500 bp) and identifies an ortholog if any region >10% of the query coverage can be mapped to the target genome, whereas IPP projections are only defined by their distance to anchor points. To test whether IC regions simply represented regions of lower sequence alignability, we mapped mouse CREs to chicken using lower LiftOver thresholds. However, LiftOver thresholds below 10% did not substantially improve ortholog detection (Extended Data Fig. 3a), indicating that IPP-unique matches are not detectable through pairwise alignments.

Other alignment-based efforts to detect orthologs include hierarchical multiple-genome alignments guided by evolutionary relationships[25,26]. We compared IPP with halLiftover/HALPER[26] for all placental mammals in our collection (Extended Data Fig. 3c). IPP performed similarly or better at relatively short evolutionary distances within placental mammals, suggesting that orthologs can be traced by comparing hundreds of genome sequences. However, IPP achieved comparable detection rates using only 16 species and spanned far greater evolutionary distances.

Finally, we tested whether IPP projected to evolutionarily conserved regions in the chicken genome by comparing their phastCons77way scores (Fig. 2d). DC projections showed the highest phastCons scores, with higher variability in enhancers than promoters. Despite lacking direct mouse–chicken alignability, IC projections had higher phastCons scores than NC, indicating that IPP projections identify evolutionarily conserved regions in the target genome without relying on sequence homology.

## Reanalysis of published CREs increases putative orthologs

As IPP could project any set of genomic coordinates, we next used IPP to project mouse CREs from studies aiming to find conserved ortholog regulatory regions: murine heart enhancers[6] and a set of limb CREs[39] represented developmental enhancers. Liver CREs[5] and a set of CEBP/A TFBS[7] were regulatory regions determined in adult livers, which might

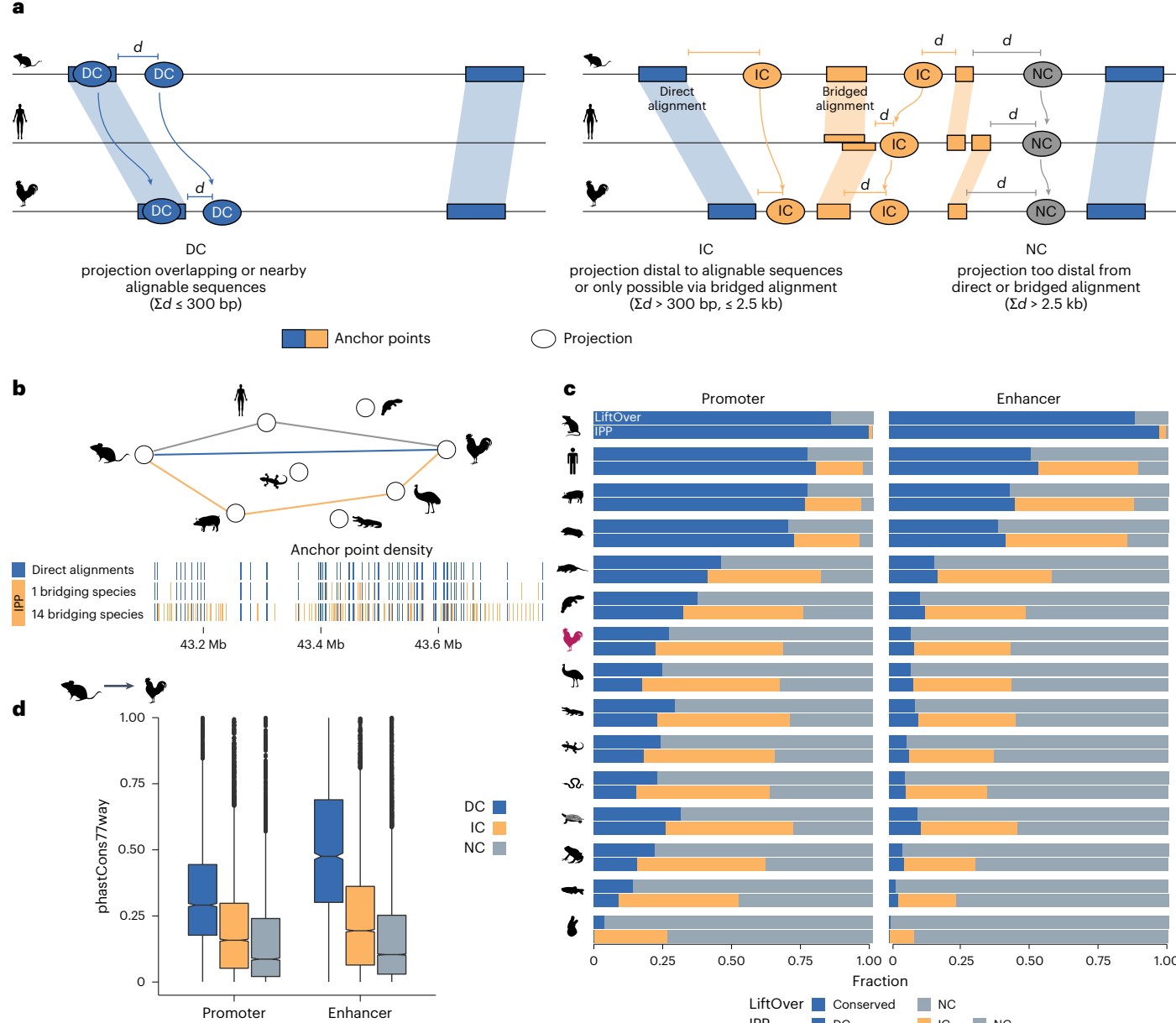

**Fig. 2 | A synteny-based algorithm, IPP, identifies thousands of putative sequence orthologs of mouse heart CREs. a**, Schematic of the IPP algorithm and its classification of DC, IC and NC features. **b**, Increase in number of available anchor points in a representative region of the mouse genome using 0, 1 and 14 bridging species. **c**, IPP increased the number of putative orthologs from mouse in 15 other species used as bridging species (compare blue versus orange portion). LiftOver alignments (top bar) are compared with IPP DC and IC alignments. The increase was particularly high at greater evolutionary distances from nonmammalian species. **d**, PhastCons77way scores for IPP-defined classes promoters and enhancers. Boxplot shows median and interquartile range of scores of 500-bp windows centered by IPP projections in chicken. Promoter n = 4,461 DC, 9,237 IC, 6,532 NC; enhancer n = 2,588 DC, 10,162 IC, 16,712 NC. **d**, distance to anchor point.

be under different selective pressures. IPP increased the number of orthologs in all datasets, equivalent to the increase seen in our heart data (Extended Data Fig. 3d–g). Heart enhancers were slightly less well conserved (8.8% DC versus 43.3% DC + IC) than limb enhancers (13.3% DC versus 49.8% DC + IC), confirming trends reported previously[6], while showing slightly higher conservation than adult liver enhancers (6.2% DC versus 38.9% DC + IC; Extended Data Fig. 3). For CEBP/A, only 2% of murine peaks were reported to be conserved in chicken, and even fewer were bound by CEBP/A in chicken livers[7]. We reanalyzed the chromatin immunoprecipitation followed by sequencing (ChIP–seq) data from mouse and chicken livers and confirmed that only a small fraction (5.7%) of mouse CEBP/A binding sites qualified as DC

in chicken, and just 173 of these sites overlapped with a CEBP/A peak in chicken (Extended Data Fig. 2f). However, by including IC projections, we increased the number of positionally conserved CEBP/A sites to 32% and found an additional set of 579 TFBS that were also CEBP/A bound in chicken livers.

Together, these results show that IPP dramatically increases detection of ortholog genomic regions independent of sequence homology, particularly for larger evolutionary distances. Although IPP projections encompass sequence-homologous regions as detected by alignment-based methods, IPP also uncovers a previously hidden set of conserved elements that can be investigated for their role in evolution and gene regulation.

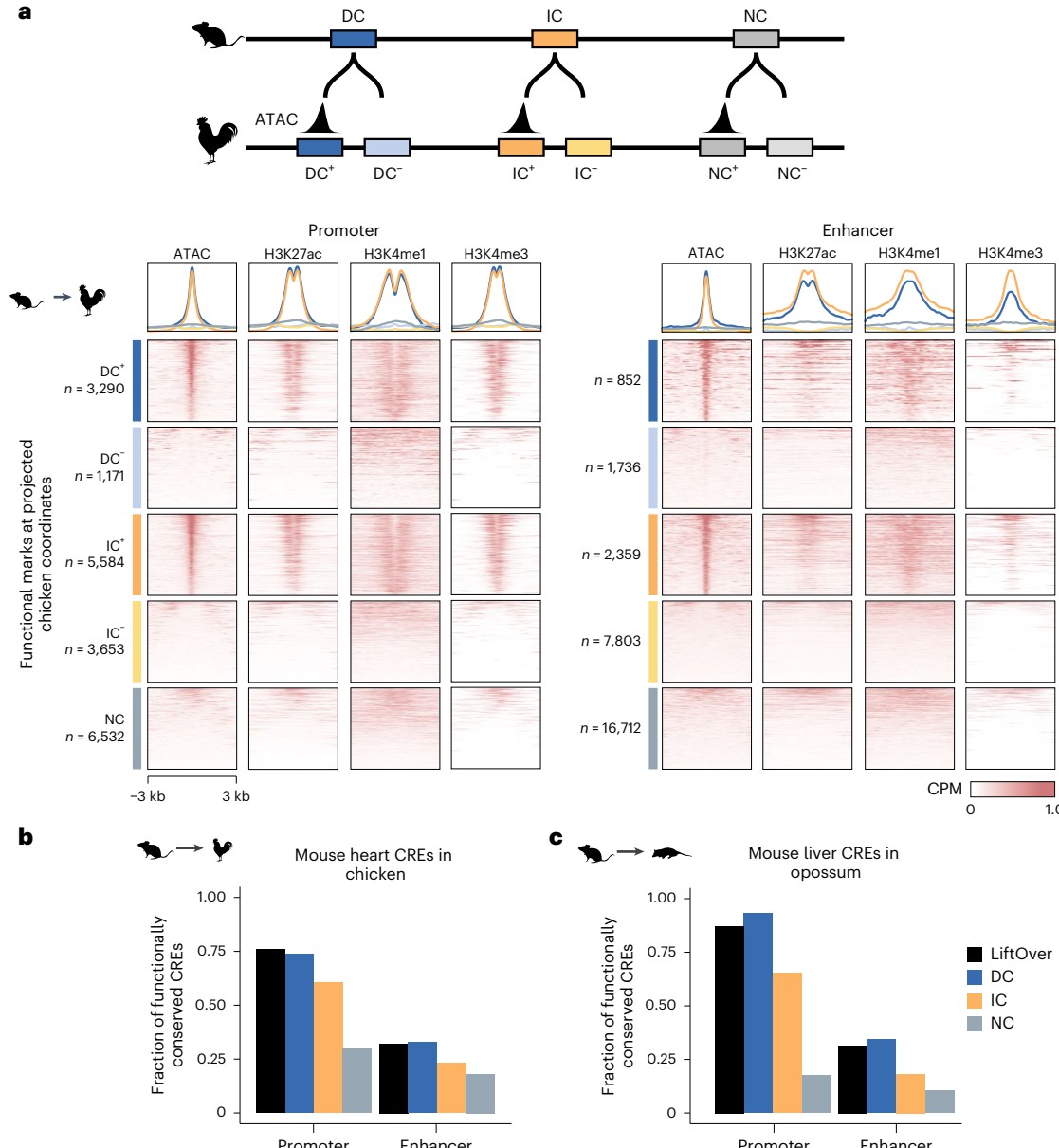

**Fig. 3 | Projections of IC and DC CREs show similar enrichment of functional chromatin marks in the chicken genome. a**, Classification of elements with or without conserved activity (±). Signal enrichment at chicken genomic regions to which mouse CREs were projected. **b**, Fractions of mouse CRE orthologs (LiftOver: minMatch = 0.1; IPP: DC, IC, NC) overlapping an ATAC-seq peak in chicken HH22/24 hearts. **c**, Fraction of mouse CRE orthologs (LiftOver: minMatch = 0.1; IPP: DC, IC, NC) overlapping an H3K27ac ChIP–seq peak in opossum adult livers.

## Functional chromatin marks at IPP-projected CREs

To determine whether DC, IC and NC CREs in the mouse genome differed in their functional chromatin signatures or genomic locations, we next compared chromatin marks across these groups. The three groups showed comparable enrichment for histone modifications and chromatin accessibility (Extended Data Fig. 4a). Moreover, analysis of ReMap[40] data showed that DC, IC and NC promoters and enhancers were similarly bound by heart TFs (Extended Data Fig. 4b). In addition, the distances of DC, IC and NC enhancers from the nearest transcription start site (TSS) were similar (Extended Data Fig. 4c). Differences in enhancer location relative to genic features were consistent with expectations based on the IPP algorithm and the higher sequence conservation of exons (Extended Data Fig. 4d). DC and IC enhancers were both more frequently located within genes than NC enhancers, but they were differently distributed relative to exons (63% of genic DC and 23% of genic IC enhancers overlapped exons).

The large additional number of IC regions suggests that up to 80% of conserved CREs might have gone undetected in most analyses to date. As we had collected functional genomic data from developmentally equivalent stages, we next tested whether IPP projections of mouse CREs pointed to regions of the chicken genome with enhancer-specific or promoter-specific chromatin features.

We classified projections with conserved accessibility as DC⁺IC⁺ and those without an ATAC-seq peak overlapping the projection as DC⁻IC⁻ (Fig. 3a). For DC CREs, we found that 74% mouse promoter and 33% enhancer projections overlapped an ATAC-seq peak in the chicken genome. An equivalent analysis for LiftOver orthologs showed similar percentages for conserved accessibility of promoters (76%) and enhancers (31%) (Fig. 3b). Notably, IC CREs performed similarly, with 60% of promoter and 23% of enhancer projections overlapping ATAC-seq peaks, although the absolute numbers were substantially higher than those for DC or LiftOver (Fig. 3a,b). Consistent with the

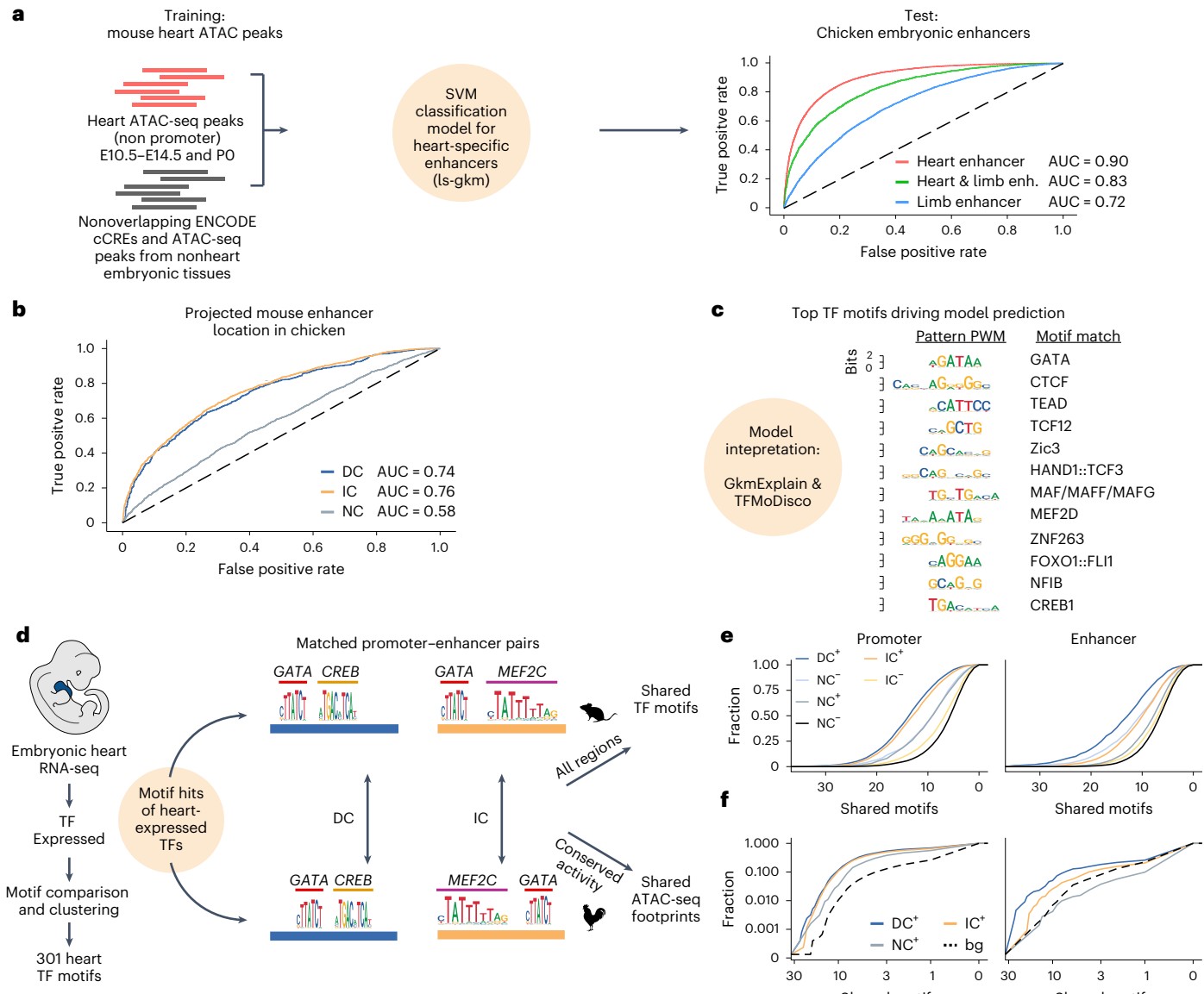

**Fig. 4 | In silico analysis of sequence composition and motif content of IC and DC elements. a**, Training of an SVM model to identify heart enhancers with independent data from public repositories. Positive set: embryonic heart or cardiomyocyte ATAC-seq peaks; negative set: nonoverlapping ATAC-seq peaks from nonheart tissues. The model distinguishes heart-specific versus limb-specific enhancers from chicken embryos. **b**, Evaluation of classification of DC⁺, IC⁺ and NC regions of the chicken genome by the SVM model. **c**, TF-MoDISco interpretation of the putative TFBS that contribute to model specificity. Binding

sites of several known heart-specific TFs contributed to model accuracy. **d**, Heart-expressed TFs identified by RNA-seq were consolidated to 301 motifs of heart-specific TFs. Promoter–enhancer pairs were screened for shared TFBS or ATAC-seq footprints. **e**, DC⁺IC⁺ promoters and enhancers shared more heart TFBS than DC⁻IC⁻ or NC regions. **f**, Functionally conserved DC and IC ATAC-seq peak pairs shared more TF footprints than NC ATAC-seq peak pairs or control pairs ('bg' indicates a nonpaired ATAC-seq peak in the same TAD).

ATAC-seq signal, acetylated histone H3 K27 (H3K27ac) was enriched around IPP DC⁺ IC⁺ projections in chicken, and there was differential enrichment of trimethylated H3K4 (H3K4me3) at promoters and monomethylated H3K4 (H3K4me1) at enhancers, suggesting that IPP IC regions identify the functional chicken orthologs of murine heart CREs (Fig. 3a).

For adult liver CREs, functional data (H3K4me3 and H3K27ac ChIP–seq) from mouse and opossum[5] enabled a comparable analysis for mouse CREs projected to opossum. We classified projections as LiftOver⁺/⁻, DC⁺/⁻ and IC⁺/⁻ based on overlap with an H3K27ac peak. We found that 93% of DC promoters and 34% of DC enhancers overlapped a H3K27ac peak in opossum, with similar percentages for LiftOver orthologs (Fig. 3c). For IC CREs, these numbers were slightly lower (65% of promoters and 18% of enhancers), but they were within a similar

range as the heart enhancer data and were consistent with previous observations for sequence-conserved DNase I hypersensitive site peaks in mammals[41]. Together, these functional chromatin marks suggest that interpolated regions point to functionally conserved CREs in the target genome and that sequence homology is an incomplete indicator of conserved activity.

## A heart-specific support vector machine model validates IPP projections

ML methods have become a viable strategy to identify cell-type-specific CREs in distantly related species without the need for sequence conservation or experimental data[18–21]. To test the regulatory potential of IPP projections in chicken, we first trained a gapped *k*-mer support vector machine (gkm-SVM)[42,43] on aggregated tissue-specific ATAC-seq peaks

from mouse embryonic heart excluding promoter regions, against a background of nonoverlapping peaks from nonheart cells and tissues (Fig. 4a, Extended Data Fig. 5 and Methods).

We then tested the cross-species predictive power of the model on our high-stringency chicken enhancers from heart and forelimb. The mouse-trained SVM correctly distinguished between heart-specific, shared and forelimb-specific chicken enhancers (Fig. 4a), confirming that sequence features learned from mouse are in fact predictive of heart-specific enhancers in chicken. A recent study found that tissue-specific CREs showed a lower degree of sequence conservation than more pleiotropic CREs[44]. We therefore assessed sequence conservation of SVM-predicted tissue specificity for all ATAC-seq peaks from chicken embryonic hearts. We observed a clear inverse relationship (Extended Data Fig. 5d), suggesting that particularly heart-specific peaks (that is, those with positive scores) are more sequence-divergent from mouse than pleiotropic peaks and providing further evidence that sequence alignability is a poor estimator of conserved regulatory activity.

We next compared the predicted tissue specificity of IPP projections of mouse enhancers in the chicken genome. DC and IC projections were equally likely to be classified as heart-specific enhancers (area under the receiver operating characteristic curve (AUC) = 0.74 (DC), AUC = 0.76 (IC)), in contrast to NC projections (AUC = 0.58) (Fig. 4b).

To better understand predictive sequence patterns learned by the model, we computed contribution scores with GkmExplain[45] and consolidated recurring high-scoring patterns (seqlets) into motifs[46]. Results from mouse and chicken largely overlapped (Extended Data Fig. 5e) and represented known motifs of master regulators of heart development (for example, GATA, TEAD and HAND), which were most predictive of a sequence being classified as a heart-specific enhancer (Fig. 4c). Thus, this independent approach confirmed that the IPP projections of mouse enhancers faithfully identified heart-specific enhancer regions in the chicken genome.

### TFBS conservation as an indicator of conserved CRE activity

If IPP projections represent positionally conserved orthologs, these pairs should share more TFBS than nonorthologous pairs. To test this, we performed TF motif scanning and ATAC-seq footprinting. We filtered our RNA-seq data to identify heart-expressed TFs and curated a set of 301 heart TF motifs (Fig. 4d). We calculated the number of shared TFBS for every mouse–chicken ortholog pair and plotted the results (Fig. 4e).

Overall, ortholog promoters shared more TFBS than enhancers. DC[+]IC[+] promoters were comparable with respect to number of shared TFBS and distinct from DC[−]IC[−] promoters (Fig. 4e). Among enhancers, DC[+] ones shared the most TFBS, whereas IC[+] enhancer pairs shared as many TFBS as DC[−] pairs. Notably, orthologs with conserved chromatin activity (dark blue and orange lines) always shared more TFBS than those without (light blue and orange lines). This suggests that functionally conserved orthologs are more likely to retain important TFBS. Finally, we compared shared TF footprints in all functionally conserved DC, IC or NC pairs relative to control ATAC peaks. To distinguish between true orthologs and other enhancers of the same gene, this control consisted of nonorthologous ATAC-seq peaks within the same TAD (Methods). Consistent with the TFBS motif scanning results, DC and IC promoters were equal with respect to number of shared TF footprints, whereas DC enhancers were slightly more likely to share TF footprints than IC enhancers (Fig. 4f). These results confirm that IPP identifies orthologous CREs with shared TFBS, representing conserved regulatory information independent of direct sequence alignability.

### IC enhancer pairs drive conserved expression patterns

Based on our analysis, both DC and IC enhancers are true orthologs. If these enhancers function as autonomous activator, they should drive conserved expression in the developing heart. To test this in vivo, we selected two pairs of DC and six pairs of IC enhancers, representing a

selection of orthologs with varying distance from target genes within TADs of known heart lineage factors (*Hand2*, *Tbx20*, *Nkx2-5* and *Gata4*) or near less well characterized heart loci (*Pakap*, *Miga1* (mm72) and *Auts2* (mm131)) (Extended Data Fig. 6a–g). Two IC pairs were smaller fragments of known mouse enhancers[47] without sequence conservation in chicken. For each of these 16 elements, we profiled enhancer activity in vivo using an LacZ enhancer–reporter integrated at a safe-harbor locus. An empty vector control produced a background signal in somites and the otic vesicle, as well as a weak outflow tract (OFT) signal in E10.5 embryos (Extended Data Fig. 7). Therefore, we considered only a LacZ signal in the ventricles or atria to indicate conserved expression.

Six of eight pairs drove conserved expression as well as boosting the OFT signal (Fig. 5a). This included both DC enhancers (*Hand2*-DC, *Tbx20*-DC) and an IC enhancer near *Gata4*. Also, three IC pairs near genes with less well described cardiac function drove conserved expression. Mouse and chicken versions of an intronic IC[+] enhancer in the *Pakap* gene (*Pakap*-IC), encoding A-kinase anchoring protein 2 (Akap2)[48], drove broad cardiac expression. Two IC[+] pairs, *Miga1*-mm72 and *Auts2*-mm131 IC[+], were 500-bp fragments of known enhancers (2.7 kb and 1.2 kb). The mouse and chicken *Miga1*-mm72 fragments recapitulated previously described ventricle-specific expression. The mouse *Auts2*-mm131 fragment boosted OFT expression with additional activity in the right ventricle and developing ear (arrowheads, Fig. 5a), whereas the chicken ortholog recapitulated OFT and ventricle expression but not ear expression. Together, these results show that IPP can faithfully identify putative orthologs in vivo despite the absence of sequence conservation.

### IC CREs show higher TFBS shuffling

In all our analyses, IC projections showed similar functional conservation to DC despite lack of alignability, raising the question of how the underlying DNA sequences may differ in the ways they encode regulatory information. We examined the enhancer pairs validated in vivo by comparing SVM-model contribution scores with predicted binding sites of key TFs (Fig. 5b (shaded boxes) and Extended Data Fig. 8) for mouse and chicken ortholog pairs.

From these initial observations, we hypothesized that for enhancer pairs with similar numbers of shared TFBS, DC pairs would display a more conserved TFBS order within the element than IC pairs (Fig. 5c). For example, for enhancer pairs with seven shared TFBS, IC pairs would show a more shuffled motif order (from 5′ to 3′) than DC pairs, probably complicating sequence alignment. To systematically test this hypothesis, we calculated the Kendall tau rank distance ($K_d$) for all enhancer pairs. This metric assesses similarity between two ranked lists by measuring the number of transpositions needed to change the order of one list into the other[49]; the more similar two lists, the smaller the distance. We selected all functionally conserved enhancer pairs with at least six shared TFBS and computed the normalized $K_d$ for each pair (Fig. 5c,d). DC enhancers exhibited significantly lower $K_d$ scores (median = 0.27) than IC (median = 0.33) and NC (median = 0.33) enhancers. Consequently, conservation of an element's regulatory function is likely to be less dependent on exact sequence conservation than on preservation of the appropriate balance of TFBS within the given element.

## Discussion

Here, we have shown widespread positional conservation of regulatory elements in the absence of sequence conservation. By combining equivalent functional genomic data, a synteny-based algorithm and in vivo validation, we revealed a substantial number of previously hidden IC elements between mouse and chicken.

Identification of orthologous enhancers is an inherently difficult problem owing to rapid enhancer evolution[2,5]. Although there have been several reports dissecting individual enhancers conserved in function rather than in sequence[8,10,12,13,22], a systematic evaluation and quantitative appraisal of this phenomenon is challenging, as it requires

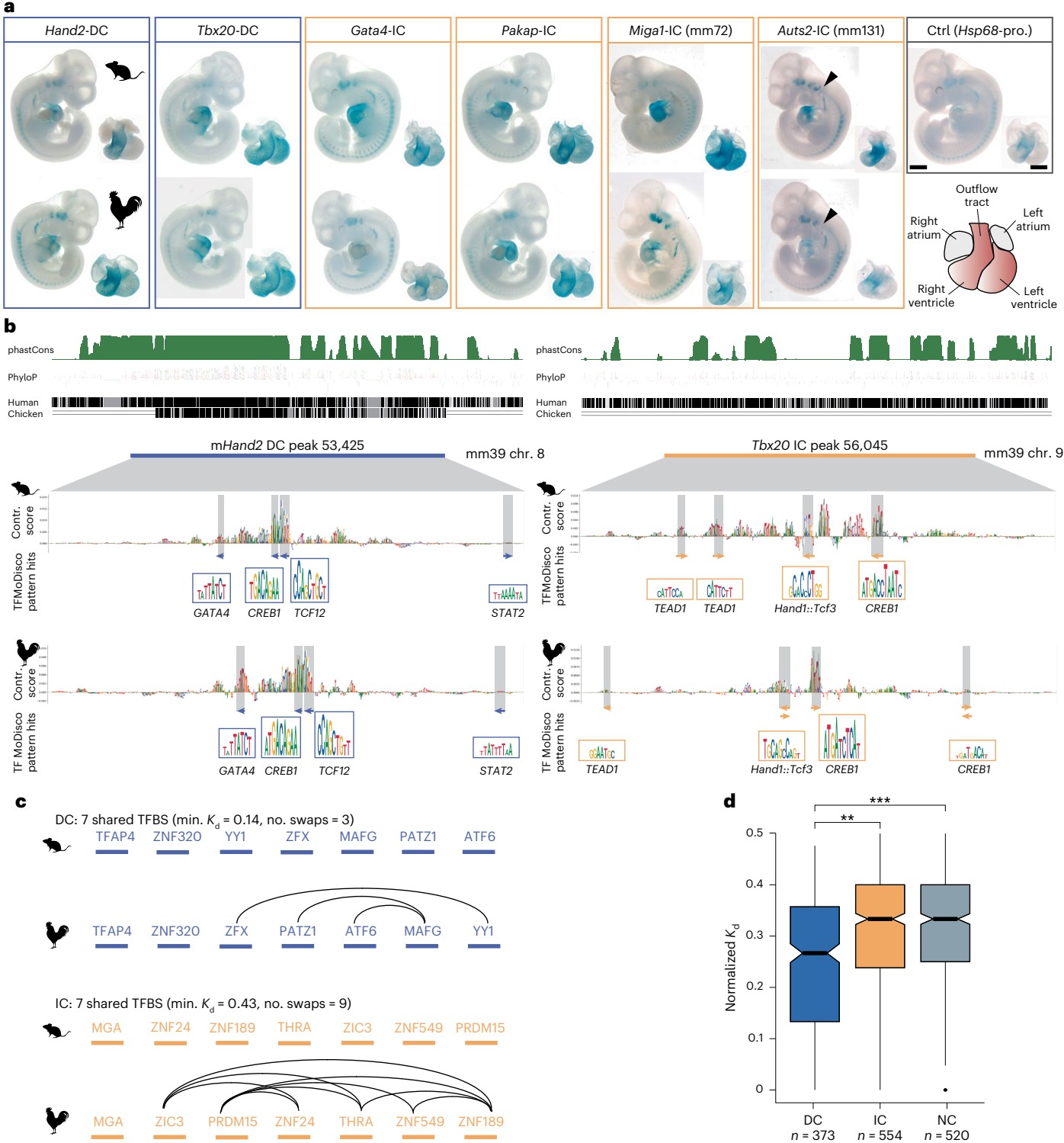

**Fig. 5 | IC heart enhancers from mouse and chicken drive conserved gene expression patterns in vivo. a**, DC and IC enhancers from mouse (top) and chicken (bottom) drive highly similar expression patterns in the hearts of E10.5 embryos. Individual enhancers show similar tissue-restricted or broad expression patterns. Scale bars, 1,000 μm (embryo), 500 μm (heart). **b**, Sequence conservation scores (phastCons/PhyloP) and direct alignments to human and chicken of the mouse *Hand2*-DC and *Tbx20*-IC enhancers tested in **a**. SVM contribution (Contr.) scores and TF-MoDISco motif matches show conserved sequence features of the 500-bp enhancer highlighting shared TF motif hits overlapping with seqlets. **c**, The different order of shared TFBS in IC and DC enhancer pairs is reflected by the computed Kendall tau distance, $K_d$. **d**, $K_d$ scores for all functionally conserved DC, IC and NC CRE enhancer pairs. Boxplot shows median and interquartile range. Asterisks indicate the magnitude of the effect size based on Cohen's $d$ (*$d < 0.2$, small; **$d \leq 0.5$, medium); $n$, number of enhancer pairs.

both alignment-free algorithmic approaches for detection and functional data for validation of such orthologous enhancers. By combining the synteny-based IPP algorithm with matching experimental data from two species, we were able to predict a large set of IC elements and demonstrate their functional equivalence to sequence-conserved elements. Our reanalysis of published data showed that these are likely to underestimate the number of chicken-conserved enhancers by fivefold[6,7]. Although our analysis did not change the general trends observed in these studies, the degree of underreported conserved regulatory elements changes the interpretation of the results with respect to which degree enhancers may evolve from neutral sequences[50] and to what degree they are conserved. Our results reveal evolutionary conservation invisible to alignment-based measures and reconcile the apparent contrast between divergent noncoding genome sequences and conserved features such as 3D chromatin structure and gene expression.

Rapidly diverging regulatory DNA presents a major challenge for studies tracing the evolution of regulatory elements across species. Multiple sequence alignments and alignment-free algorithms can identify orthologs, but this is especially challenging for large evolutionary distances. Hierarchical alignments such as halLiftover/HALPER[25,26] require multiple alignment of hundreds of genomes but perform similarly to IPP using only 16 genomes; this highlights the potential of synteny as a proxy for conservation. Bridged or tunneled alignments have already indicated a higher degree of CRE conservation than commonly assumed[23,38]. IPP builds on this and extends it in several ways. First, IPP implements multiple bridging species, which can be optimized for pairwise comparison based on their specific phylogenetic relationships. Second, within the framework of conserved synteny, IPP assumes orthology for any pair of genomic positions between any two genomes, irrespective of DNA sequence. Consequently, in nonsyntenic regions or very distantly related genomes[14], IPP might miss orthologous elements. Nevertheless, IPP is a potent approach to identification of putative orthologs that can be used in comparative studies at varying evolutionary distances. Especially when combined with equivalent experimental data, as in our study, IPP can drastically increase the number of conserved orthologs compared to sequence conservation. As such, IC elements can provide valuable information for human disease-associated noncoding variants and their functional characterization in animal models, for example, in congenital heart disease[51,52].

Advances in ML have enabled prediction of cell-type-specific regulatory activity for any DNA sequence[53-57]. Within mammals, models trained in one species can successfully predict activity in another[18,20,21] but cannot match ortholog pairs. A recent study that aimed to identify human and mouse orthologs used an ML model first but relied on syntenic blocks to match orthologs[19]. Here, we showed that our murine SVM model also predicted heart-specific enhancers in chicken and used it to validate IPP projections. In the future, combinations of ML models and IPP could represent a powerful strategy for study of enhancer evolution. For example, IPP-identified ortholog pairs could serve as training input for ML models to learn sequence changes compatible with functional conservation.

Enhancer sequence conservation ranges from ultraconserved elements[58-60] to the sequence-divergent IC elements we describe. Notably, functional conservation in chromatin state (that is, DC$^+$IC$^+$ elements) accounts for a surprisingly small fraction, in spite of our efforts to assess high-confidence CREs from stage-matched tissues. This is true not only for IPP projections; it also holds for LiftOver orthologs and was consistent in adult liver enhancers[5] reanalyzed here, as well as being reported for sequence-conserved DNase I hypersensitive sites in equivalent tissues obtained in the ENCODE consortium[41]. This suggests that alternative activity of ortholog enhancers might be more widespread than currently appreciated, and functional conservation—in terms of chromatin signatures, encoded TFBS and predicted tissue specificity—is relatively uncoupled from sequence conservation. In this

light, we expect IPP to be an efficient method of increasing the number of functionally conserved ortholog CREs between species, for example, in single-cell ATAC-seq or ChIP–seq datasets from equivalent tissues, in which cell types and expression programs are conserved, whereas the majority of CREs currently appear to be NC.

Finally, the TFBS shuffling analysis suggested that conservation of an element's regulatory function is less dependent on exact sequence syntax than on preservation of the appropriate balance of TFBS within the given element. Given that we found thousands of IC elements between mouse and chicken, the functional conservation of CREs across larger evolutionary distances is likely to be much more prevalent than currently appreciated.

## Online content

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

## Methods

### Ethics and consent

This study complied with all relevant ethical regulations. Animal experiments complied with German Animal Welfare Law (TierSchG) and received approval from the local authorities (LaGeSo Berlin G0098/23).

### Biological samples

C57BL/6 inbred mice were used for timed mating, and fertilized specific-pathogen-free eggs (Valo Biomedia) were incubated at 38 °C, 50–55% humidity. Embryonic hearts and forelimbs from mouse and chicken embryos (E10.5 and E11.5, and HH22 and HH24) were dissected and processed for functional genomic assays.

### Preparation of samples and sequencing libraries

**RNA-seq.** Dissociated chicken embryonic heart cells were snap-frozen. RNA extraction (Qiagen RNeasy-Mini Kit) was followed by ribosomal RNA depletion and then library preparation (Kapa HyperPrep Kit).

**ATAC-seq.** For the Omni-ATAC protocol (50,000 cells per replicate), embryonic tissues were dissociated into a single-cell suspension, washed with cold phosphate-buffered saline (PBS) and lysed in fresh lysis buffer (10 mM TrisCl pH 7.4, 10 mM NaCl, 3 mM MgCl$_2$, 0.1% (v/v) Igepal CA-630) on ice. Tn5 transposition was performed for 30 min at 37 °C, followed by DNA purification (MinElute Reaction Cleanup kit, Qiagen).

Nextera indexing primers were added during library amplification (amplification cycles determined by quantitative PCR), followed by double-sided size selection and nucleosomal fragment distribution validation (BioAnalyzer, TapeStation). Library concentration was measured with Qubit.

**ChIPmentation.** For ChIPmentation, following the protocol described previously[61], dissociated cells were filtered through a 100-µm (embryonic heart) or 70-µm (limb) MACS SmartStrainer before fixation (1% methanol-free formaldehyde (Thermo Scientific: 28906) in PBS) for 10 min on ice. Fixation was quenched using glycine, and cells were lysed in lysis buffer (10 mM Tris pH 8.0, 100 mM NaCl, 1 mM EDTA pH 8.0, 0.5 mM EGTA, 0.1% sodium deoxycholate, 0.5% N-lauroylsarcosine) before sonication (Covaris E220, fragment distribution 200–700 bp). Antibody incubation (1 µl per ChIP) was performed overnight at 4 °C, followed by immunoprecipitation (protein G beads). After washing, Tn5 tagmentation was performed at 37 °C for 5 min. Beads were washed again and then subjected to overnight reverse crosslinking with Proteinase K. DNA was purified using MinElute Reaction Cleanup kit (Qiagen).

Libraries were indexed and amplified similarly to ATAC-seq libraries (amplification cycles per library determined by quantitative PCR $C_t$ values (number of cycles = rounded up $C_t$ value + 1). After amplification, DNA was purified with AmPureXP beads, validated using a TapeStation D5000HS and subjected to size selection. Final DNA concentration was measured (Qubit HS) and again validated (TapeStation D5000HS).

**Hi-C.** For in situ Hi-C library preparation[62], DpnII digestion of fixed cells was followed by biotin-14-dATP incorporation. DNA was sheared (S-Series 220 Covaris) to 300–600-bp fragments before biotin pull-down (Dynabeads MyOne Streptavidin T1 beads). DNA end repair was performed with T4 DNA polymerase and a Klenow fragment, followed by phosphorylation with T4 polynucleotide kinase. Sequencing adapters were added, and libraries were indexed via PCR amplification (4–8 cycles) using NEBNext Ultra II Q5 Master Mix. PCR clean-up was then performed, followed by AmPureXP size selection.

### Data processing

**RNA.** RNA-seq data were processed with STAR v.2.7.9a using reference genome sequences and annotations from GENCODE (v.M32, primary) for mouse and Ensembl (GRCg7b) for chicken. We obtained gene-level counts with --quantMode geneCounts. In addition to in-house chicken heart RNA-seq libraries, we similarly processed the following publicly available datasets: mouse heart E10.5 and E11.5 (ENCODE3); and chicken forelimb HH22 and HH24 (GSE164737)[63]. TPM (transcripts per million) values were computed from gene-level counts; gene length was estimated as the sum of all exon lengths.

**ATAC-seq and ChIPmentation.** For ATAC-seq and ChIPmentation, Nextera Tn5 adapter sequences were trimmed from fastq reads using cutadapt. Then, reads were aligned to reference genomes (mm10, mm39 or galGal6) using bowtie2 v.2.3.5.1 with the maximum fragment length set to either 1,000 bp (ATAC) or 700 bp (ChIPmentation). Duplicated reads were removed using MarkDuplicates (Picard v.2.23.4). Finally, reads were sorted and filtered using samtools v.1.10 to remove unmapped, low quality (mapping quality < 10) and mitochondrial reads. Filtered bam files from replicates were merged to generate bigwig files using bamCoverage (deepTools) with counts per million normalization and bin size of 1 (ATAC) or 10 (ChIPmentation). ATAC-seq peaks from replicates were called with Genrich v.0.6.1 in ATAC mode '-j' using default parameters (https://github.com/jsh58/Genrich).

**Hi-C.** Reads were handled using Juicer v.1.6.0 CPU version[64], specifically aligned using BWA-MEM v.0.7.17 to reference genome galGal6. Only read pairs with mapping quality > 30 were included in the final contact maps. Processing was performed separately for each replicate, and output filtered deduplicated read pairs were merged. Contact matrices were Knight–Ruiz normalized[65] before visualization.

### Data analysis

**Comparative differential expression analysis.** Raw gene-level counts from heart and limb samples at both stages were used as input for DESeq2 (v.1.36)[66]. We obtained a set of differentially expressed genes in the heart relative to limb in both stages, accounting for the effects for biological replicates. To aid visualization and gene ranking for gene ontology analysis, effect size shrinkage was performed for the coefficient modeling tissue specificity (that is, tissue_heart_vs_limb).

Gene orthology annotations were obtained from Ensembl databases GRCm39 (mouse) and GRCg7b (chicken). Duplicate annotations were filtered, retaining those with the highest gene order conservation scores. Only one-to-one orthologous genes were used for comparative analyses. Gene ontology analysis was done using R package clusterProfiler (v.4.4.4)[67]. Overrepresentation gene ontology analysis of orthologous genes was done given a background gene set of all detectably expressed mouse genes (that is, raw counts ≥ 10). For statistical testing, the sizes of test gene sets were set from a minimum of 5 to a maximum of 100 genes to enable a greater focus on specific biological processes than on more general terms.

**Estimation of sequence alignability.** To estimate conservation by means of sequence alignability, we used UCSC LiftOver as implemented within R package rtracklayer for reciprocal mapping between mouse and chicken genomes. Chain files for mm39 and galGal6 were obtained from UCSC and imported into R using rtracklayer. For mapping, we used default parameters (minMatch = 0.1) and allowed multiple (one-to-many) mapping between query and target.

**Enhancer and promoter prediction.** H3K27ac, H3K4me1 and H3K4me3 histone profiles (merged replicates) were used as input for CRUP[37]. CRUP computes the probability of being an active regulatory element for each 100-bp bin. In combination with normalized histone signal values (mono/tri ratios), bins were filtered and merged into promoter- or enhancer-like regions.

Promoters were defined by intersecting CRUP-defined promoter-like regions with all TSS of transcribed genes (counts ≥ 1 TPM; described above). Next, we filtered the set of active enhancers by their

accessibility, as determined by ATAC peaks. Finally, enhancers within 2 kb of a predicted promoter were removed from the final set. The numbers of enhancers and promoters can be found in Supplementary Table 1 and the bed files under accession codes GSE263587, GSE263753, GSE263755 and GSE263783.

**Hi-C analysis of GRBs.** *Identification of CNEs*. CNEs between chicken (*Gallus gallus*, galGal6) and mouse (*Mus musculus*, mm10) were identified using pairwise axt net whole-genome alignments downloaded from the UCSC Genome Browser. The alignments were processed using the CNEr package[68] (v.1.40.0) in R. Regions with at least 70% sequence identity over 50 base pairs were considered CNEs. We discarded elements aligned to the genome using BLAT more than four times. We calculated the CNE densities by smoothing the distribution of CNEs across the genomes using sliding windows. We used a window size of 300 kb for mouse and 100 kb for chicken.

*Identification of GRBs*. We identified GRBs as regions with a high density of syntenic CNEs, as previously described[33]. In brief, we applied an unsupervised two-state hidden Markov model to the smoothed CNE density profiles to partition the genome into regions of high and low CNE density. We excluded CNEs located outside the high-density regions from further analysis.

Adjacent CNEs within the high-density regions were merged based on their distances. We set a threshold at the 98th percentile of the gap distribution to split the genome into discrete regions. We used this threshold, previously determined[33], between human and chicken genomes. Regions were further divided based on synteny information to generate discrete syntenic blocks. Regions lacking protein-coding genes were merged with adjacent regions if they were within 300 kb. We discarded any regions that had fewer than ten CNEs.

*Hi-C data processing and visualization*. We obtained Hi-C interaction data in .hic format and converted them to multiresolution cooler (.mcool) files using HiCExplorer[69] v.3.6. This conversion facilitated access to Hi-C interaction matrices at various resolutions for downstream analysis. We calculated the directionality index (DI) using FAN-C[70] v.0.9.28. We generated funnel plots using the Genomation package in R v.1.36.0.

For both species, GRBs were ordered, centered and uniformly extended to match the size of the largest GRB in the dataset. Each GRB was sliced into 500 bins, and the average DI was calculated for each bin. The binarized DI heatmaps show the average DI in each bin converted to a binary value: bins with a positive average DI are assigned a value of 1 (red), and bins with a negative average DI are assigned a value of −1 (blue). This binary representation highlights regions with a pronounced directional bias in chromatin interactions. The funnel shape observed in the heatmaps indicates that GRB boundaries correspond to changes in interaction bias toward the interior of the GRB.

**TFBS motif and footprinting analysis**
**Reference motif collection.** We obtained TF motif models from JASPAR 2022 (core vertebrate, nonredundant) and systematically curated this database for TFBS-based analysis. From more than 700 JASPAR TF motifs, we filtered for those TFs with detectable expression in mouse embryonic heart by integrating RNA-seq counts (described above) of ≥1 TPM in both replicates, in either stage E10.5 or E11.5 (*n* = 520). From these, we consolidated the reference collection by filtering out redundant motifs based on sequence similarity within annotated TF families. Specifically, within each TF family, motifs were ranked by informational content score before pairwise comparison with others in the family using compare_motifs from R package universalmotif. Finally, motifs with lower informational content score and similarity score > 0.9 (a score of 1 indicates an identical sequence) were discarded from the final reference set (*n* = 301).

**Motif scanning.** To characterize TFBS composition, we searched matches to the 301 reference motifs using FIMO (R package memes[71]) with default parameters. CRE DNA sequences were obtained from BSgenome.Mmusculus.UCSC.mm39 and BSgenome.Ggallus.UCSC. galGal6 for mouse and chicken, respectively. Motif scanning was done within a 500-bp window centered by ATAC peak summit or projected point. Peak centering by summit was done for projected regions in chicken for functionally conserved elements (that is, DC$^+$ and IC$^+$). Any overlapping hits from the same motifs were discarded, keeping the match with higher score.

**ATAC-seq footprinting.** Aligned ATAC-seq reads from biological replicates were merged as input for ATAC-seq footprinting using TOBIAS[72] (v.0.3.3). Footprinted regions were: (1) the union set of predicted enhancers and promoters in mouse and chicken; and (2) all called chicken ATAC-seq peaks. Briefly, we corrected for Tn5 bias before calculating footprint scores at genomic regions of interest. We used our curated set of TFBS motifs as a reference to predict TF binding. TOBIAS outputs from different stages were merged, and overlapping regions of predicted binding from the same TF were merged similarly to motif hits as described. Quantification of shared footprints was done as for the motifs analysis.

**Quantification of motifs and TFBS sharing between pairs of orthologous CREs.** Similarity between mouse CREs and IPP-defined chicken orthologs was quantified as follows: we determined the total number of shared motifs and TFBS between every mouse–chicken pair. As a negative control, we compared the number of shared motifs between a mouse sequence and a nonorthologous (background) region. For every mouse sequence with a chicken projection overlapping an ATAC-seq peak (that is, DC$^+$IC$^+$NC$^+$), another ATAC-seq peak (if possible, within the same TAD) was randomly selected as its nonortholog.

**Classification model for heart-specific enhancers**
**Training strategy and data preparation.** Our classification model was an SVM with a center-weighted radial basis gapped *k*-mer kernel function (wrbfgkm) (implemented at https://github.com/kundajelab/lsgkm-svr)[42,43]. All datasets used for model training were bulk ATAC-seq peaks obtained either from ENCODE or in-house (as described above). To learn predictive features of heart-specific enhancers, we constructed the positive set to include called ATAC-seq peaks from mouse hearts at six developmental stages (in-house: E10.5 and E11.5; ENCODE: E12.5–E14.5 and P0), centered at the peak summit and extending 250 bp on either side. To exclude promoters, regions within 2 kb of annotated mouse promoters (from the EPD3 database) were removed from the final training set (*n* ≈ 65,000).

For model training, we constructed the negative set such that the model learned sequence features determining whether an enhancer or CRE was heart-specific. First, to limit confounding factors, we generated a tenfold null set from random genomic loci; then, we filtered for those overlapping any annotated ENCODE candidate CREs or ATAC-seq peaks from five nonheart embryonic organs (limbs, midbrain, forebrain, hindbrain and liver; E12.5) and mouse embryonic stem (mES) cells. Finally, those within 2 kb of any region from the positive set were removed (*n* = 70,000). All negative sets of GC- and repeat-matched sequences were generated using the genNullSeqs function from R package gkmSVM[42,73]. Repeat-masked genomic sequences were obtained from custom masked BSgenome data packages for mm10, mm39 or galGal6.

**Hyperparameter tuning and performance evaluation.** Classification performance was measured by AUC. For parameter tuning, a grid search for the *C* and *g* parameters for wrbfgkm-kernel was done using a fivefold cross validation for each combination of *C* = 1, 5, 10, 20 and *g* = 0, 1, 2, 5 (16 conditions). The best-performing parameter set (*C* = 10, *g* = 2) as

determined by its calculated AUC was chosen for model training. The final model was tested on positive versus negative regions on held-out chromosomes 1 and 2.

**Model prediction on chicken CREs and projections.** Our heart enhancer SVM model trained on mouse sequences was used to classify: (1) identified chicken enhancer and promoter sequences from heart versus forelimb; and (2) mouse CRE-projections in the chicken genome. For each prediction, the negative set generated as described previously consisted of GC- and repeats-matched regions. In addition, only projected regions overlapping ATAC-seq peaks (that is DC$^+$ or IC$^+$) were included in the analysis. AUCs were computed to evaluate the model's performance on these regions.

**Model interpretation and de novo motif discovery.** We used Gkm-Explain[45] (implemented at https://github.com/kundajelab/lsgkm-svr) to interpret model classification. GkmExplain computes contribution scores at each nucleotide in all input sequences, that is, the importance score of the sequence. For each sequence, this importance score was computed by element-wise multiplication of the one-hot encoded sequence matrix by its hypothetical importance score. Scores were visualized using the visualization module from Python package modisco.

Computed hypothetical scores were normalized by the ratio of original importance scores and sum of all hypothetical scores having the same sign. Normalization better reflected the importance score of a specific base at each position, reducing noise for subsequent motif discovery with TF-MoDISco[46] (implemented at https://github.com/jmschrei/tfmodisco-lite). Normalized scores from GkmExplain from the (1) mouse positive test set ($n = 9,000$) and (2) heart-specific chicken enhancers ($n = 15,000$) were used as separate inputs for TF-MoDISco runs. Similar positive sequence patterns from these runs were merged for the final set of predictive sequence patterns stored as position weight matrices. Flanking positions with information content $<0.5$ were trimmed from the position weight matrices before being annotated with TOMTOM[74] using our TF motifs collection as a reference.

**Quantification of motifs shuffling**
We measured the $K_d$ for TF motif hits between pairs of mouse and IPP-projected chicken enhancers. We considered each pair of mouse–chicken sequences as two ranked lists of motifs, where the order of shared motifs 5′–3′ represented the rank. The 5′–3′ order of motifs for mouse enhancer was the reference and was compared with both possible orientations in chicken.

To ensure that we faithfully encoded the specific order of motifs as ranks, shared motifs obtained previously were further processed to filter out largely overlapping occurrences from different motifs (minimum overlap: 8 bp), keeping the hits with the highest mapping score. In addition, to ensure unique rankings, runs of hits from the same motif were considered to be a singular match. Any sequence containing $>1$ noncontiguous hits from the same motif (for example, A,B,C,A,D) was stored as a matrix of ranking lists, in which each row represented a unique ranking order (for example, 1-A,B,C,D and 2-B,C,A,D). Using R package rankdist[49], we computed the normalized $K_d$ between all unique ranking lists for a mouse–chicken pair, which accounted for varying numbers of shared motifs (that is, list length). Finally, we took the smallest computed $K_d$ value for each pairwise comparison and compared among conservation classes DC, IC and NC. The effect size of sequence conservation on shuffling was determined by computing Cohen's $d$ using R package effsize[75].

**In vivo enhancer–reporter assays**
Transgenic mice carrying enhancer–reporter transgenes were generated using a PhiC31 system[76] for site-specific integration into recipient mES cells. Genomic regions and primers used for genotyping enhancer–reporters are listed in Supplementary Table 1.

We established a recipient mES cell line with a Hsp68::LacZ expression cassette containing the attP site at a safe-harbor locus (H11) via CRISPR–Cas9 knock-in. For enhancer knock-in, each individual enhancer was Gibson-cloned into a donor vector containing the attB site and a puromycin selection marker. Subsequently, each donor vector was cotransfected with a PhiC31 expression plasmid into recipient mES cells using Lipofectamine LTX (Invitrogen), and clonal mES cell lines were established. Transgenic embryos were generated from enhancer–reporter mES cell lines via tetraploid complementation[77] by the MPI-MG transgenic core facility. At E10.5, the embryos were harvested and processed for LacZ staining. In brief, the embryos were kept in the dark for 60 min at 37 °C in LacZ staining buffer supplemented with 0.5 mg ml$^{-1}$ X-gal, 5 mM potassium ferrocyanide and 5 mM potassium ferricyanide. After 60 min, the embryos were washed several times in PBS and kept overnight at 4 °C. Embryos were fixed with 4% paraformaldehyde and PBS supplemented with 0.2% glutaraldehyde and 5 mM EDTA for long-term storage at 4 °C. Imaging was performed with a SteREO Discovery.V12 microscope and Leica DFC420 camera. Z-stacks were generated using the PMax function in ZereneStacker (v.1.04). Embryo genotyping was performed by PCR using primers spanning the expected 5′ and 3′ integration junctions to confirm correct integration of the enhancers.

**Statistics and reproducibility**
Functional genomic assays were performed in two biological replicates. LacZ stainings show representative results of at least three independent replicates. Graphs represent the mean or median values obtained from $n$ biological replicates as indicated. Box plots indicate the median and interquartile range. Statistical tests used are reported in the figure legends. Data collection and analysis were not performed blind to the conditions of the experiments. The data distribution was assumed to be normal, but this was not formally tested.

**Reporting summary**
Further information on research design is available in the Nature Portfolio Reporting Summary linked to this article.

## Data availability
ChIPmentation, RNA-seq, ATAC-seq and Hi-C sequencing data generated from the chicken embryonic heart and forelimb and mouse embryonic heart ATAC-seq data study have been deposited to NCBI Gene Expression Omnibus (GEO) under accessions GSE263587, GSE263753, GSE263755 and GSE263783. The following published datasets were reanalyzed (details are listed in Supplementary Table 1): (GSM2544836, GSE185775 ENCSR582SPN, ENCSR266JQW, ENCSR782DGO, ENCSR782DEA, ENCSR222IHX, ENCSR963OLG, ENCSR886IHN, ENCSR592GQI, GSE164737 and GSE164738). Source data are provided with this paper.

## Code availability
The source code for IPP and the pairwise alignment files for the set of bridging species used in this study are available via GitHub at https://github.com/tobiaszehnder/ipp (ref. 78).

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

## Acknowledgements

D.M.I. and M.P. were supported by funding from the DFG SPP 22.02 '3D Genome Architecture in Development and Disease' (IB139/1-1 and IB 139/6-1). Work in the Ibrahim laboratory is supported by an ERC Starting Grant SYNREG (101076709). D.B. was funded by the European Union: NextGenerationEU grant NPOO.C3.2.R2-I1.06.0060. We thank the MPI-MG transgene facility and animal house for generation of transgenic embryos and members of D. Seelow's and M. Kircher's laboratories for feedback on the ML analysis; S. Zehnder for his contribution to the implementation of the computational framework in C++; and J. Glaser, A. Madgwick and all members of the Ibrahim laboratory for feedback on the manuscript.

## Author contributions

M.P. designed and performed most experiments, analyzed the data and performed the ML analyses. T.Z. developed the IPP algorithm with contributions from F.M., B.L. and M.V. and analyzed the data. M.V. suggested the Kendall tau analysis. F.P. established the enhancer–reporter mES cell line. Enhancer–reporter assays were performed by M.P. and D.M.I. with help from F.P., A.M., M.A., H.M.W. and B.M. D.B. performed the GRB analysis. B.-W.L. conducted the HALPER/halLiftover analyses. D.M.I. conceived the study, designed experiments, analyzed the data and drafted the manuscript. M.P. and D.M.I. wrote the manuscript with contributions from the other authors. All authors read and approved the manuscript.

## Funding

## Competing interests

The authors declare no competing interests.

## Additional information

**Extended data** is available for this paper at https://doi.org/10.1038/s41588-025-02202-5.

**Correspondence and requests for materials** should be addressed to Daniel M. Ibrahim.

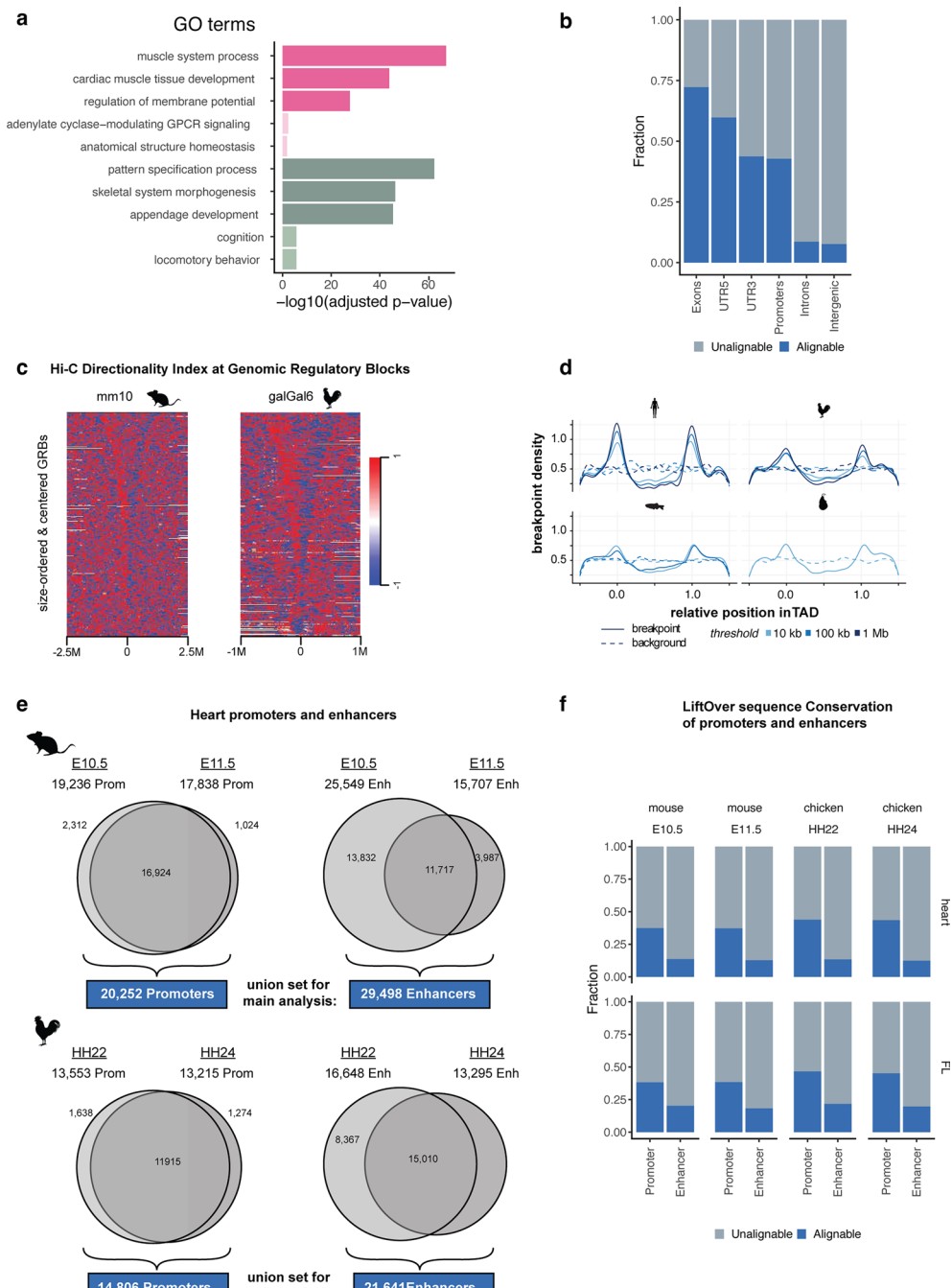

**Extended Data Fig. 1 | Conservation of gene expression and 3D chromatin structure in contrast to cis-regulatory element sequences. (a)** Gene Ontology (GO) annotations of differentially expressed genes (Heart vs. FL) in mouse and chicken. Dark pink = upregulated, both species. Dark green = downregulated, both species. Light pink = upregulated, mouse-only. Light green = upregulated, chicken-only. Grey = no differential expression. **(b)** Estimation of sequence alignability (LiftOver minMatch=0.1) of ATAC-seq peaks from mouse embryonic heart at different annotated genomic locations. **(c)** Distribution of binarized directionality indices across centred and size-ordered genomic regulatory blocks in mouse and chicken **(d)** Synteny breakpoints between mouse and human, chicken, zebrafish and *Ciona intestinalis* genomes relative to the normalized TAD position **(e)** Number of predicted promoters and enhancers from stage-specific and shared/union sets in both species. **(f)** Estimation of sequence alignability from stage-specific predicted promoters and enhancers from heart and forelimb (FL) in both species.

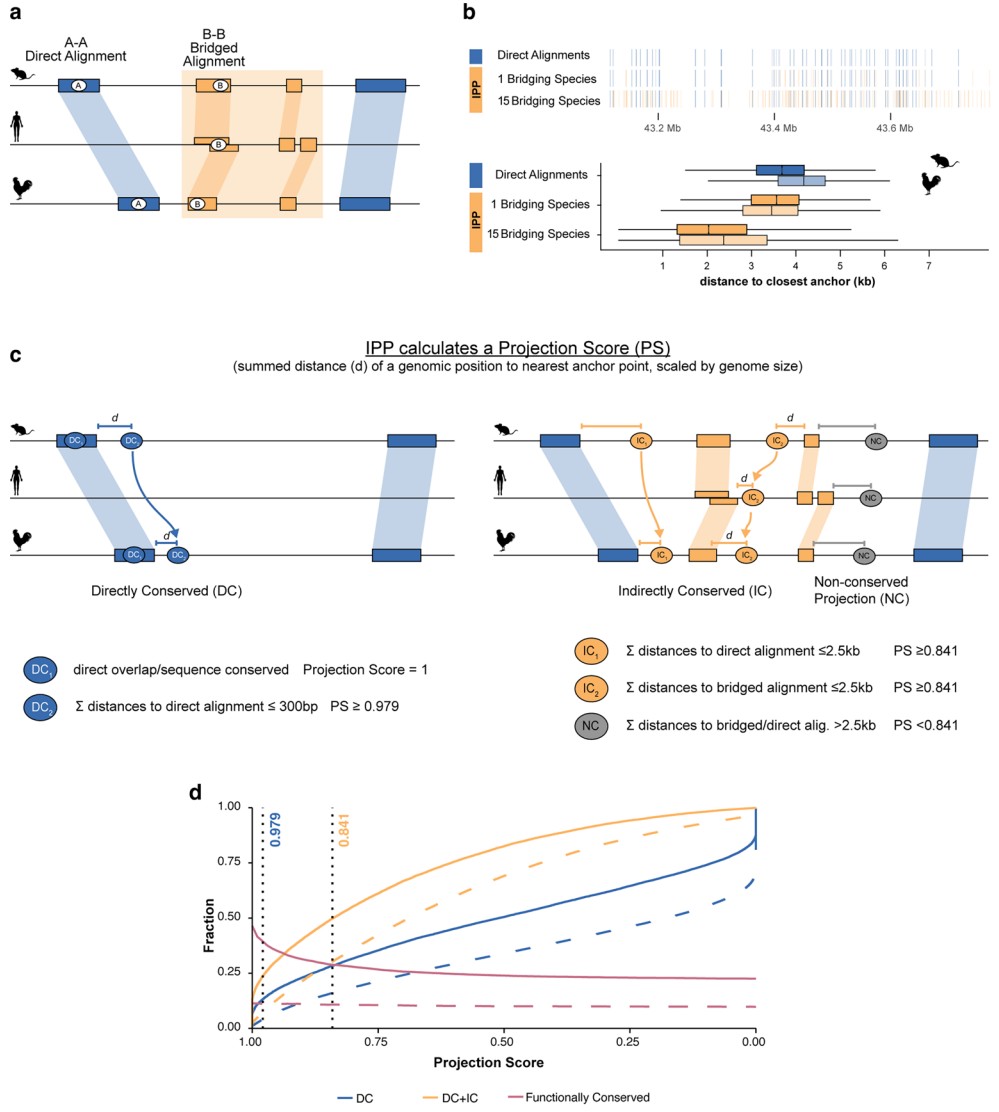

**Extended Data Fig. 2 | Interspecies Point Projection combines bridged alignments and synteny to identify orthologous regions. (a)** Classification of direct and bridged alignments through the use of intermediate species **(b)** Increase in the number of anchor points and distance to the nearest anchor points through multi-species bridged alignments. Comparison between 0, 1 and 15 bridging species **(c)** Classifcation of projections as directly and indirectly conserved. DC regions overlap a sequence alignment or are ⊡ 300 bp from a direct alignment. The distance of IC regions as >300 bp but ⊡ 2.5 kb from a direct or indirect alignment. Regions with >2.5 kb summed distance through the species graph from anchor points are classified as NC. **(d)** Fractions of mouse enhancers identified as directly conserved (DC, blue) or either directly or indirectly conserved (DC + IC, orange) as a function of the projection score threshold. Fraction of functionally conserved DC + IC elements as a function of the projection score threshold (red). Solid lines = enhancers, dashed lines = randomly selected background regions. Dotted vertical lines represent DC threshold score of 0.979 and IC of 0.841.

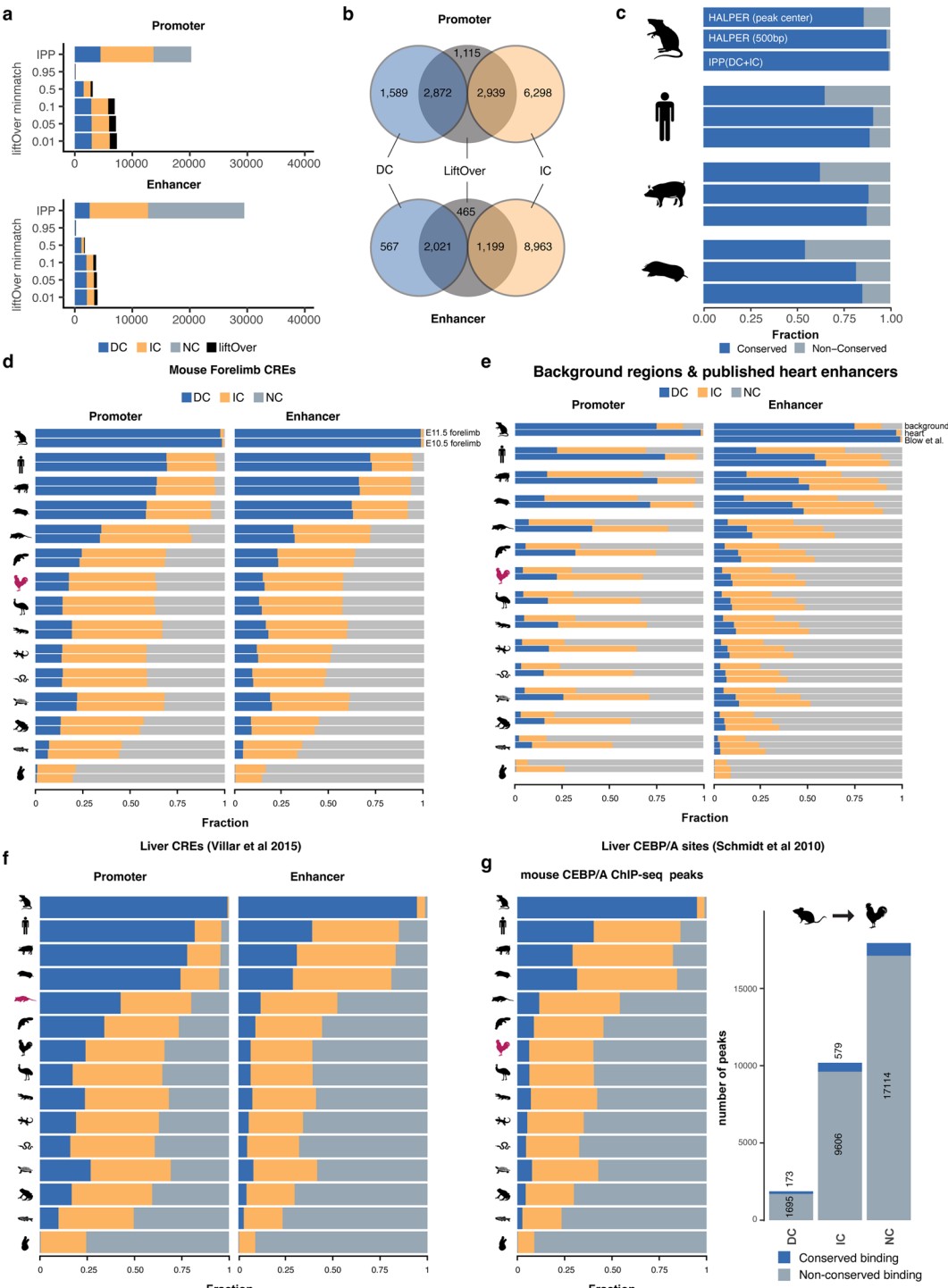

**Extended Data Fig. 3 | Interspecies Point Projection increases detection of putative ortholog regions compared to alignment-based methods.** (a) IPP projections compared to LiftOver determined orthologs using variable minMatch thresholds. (b) Overlap of DC and IC projections with LiftOver-determined orthologs (minMatch=0.1) (c) IPP performance compared to halliftover/HALPER for mouse heart enhancer ortholog prediction in four placental mammals. (d-f) IPP projections for forelimb CREs at E10.5 & E11.5 (d), randomly selected genomic regions and published heart enhancers (Blow et al 2010) (e), adult liver CREs (f). (g) IPP projections for mouse CEBP/A ChIP-seq peaks and number of conserved binding events (as determined by overlap with a CEBP/A ChIP-seq peaks in chicken livers).

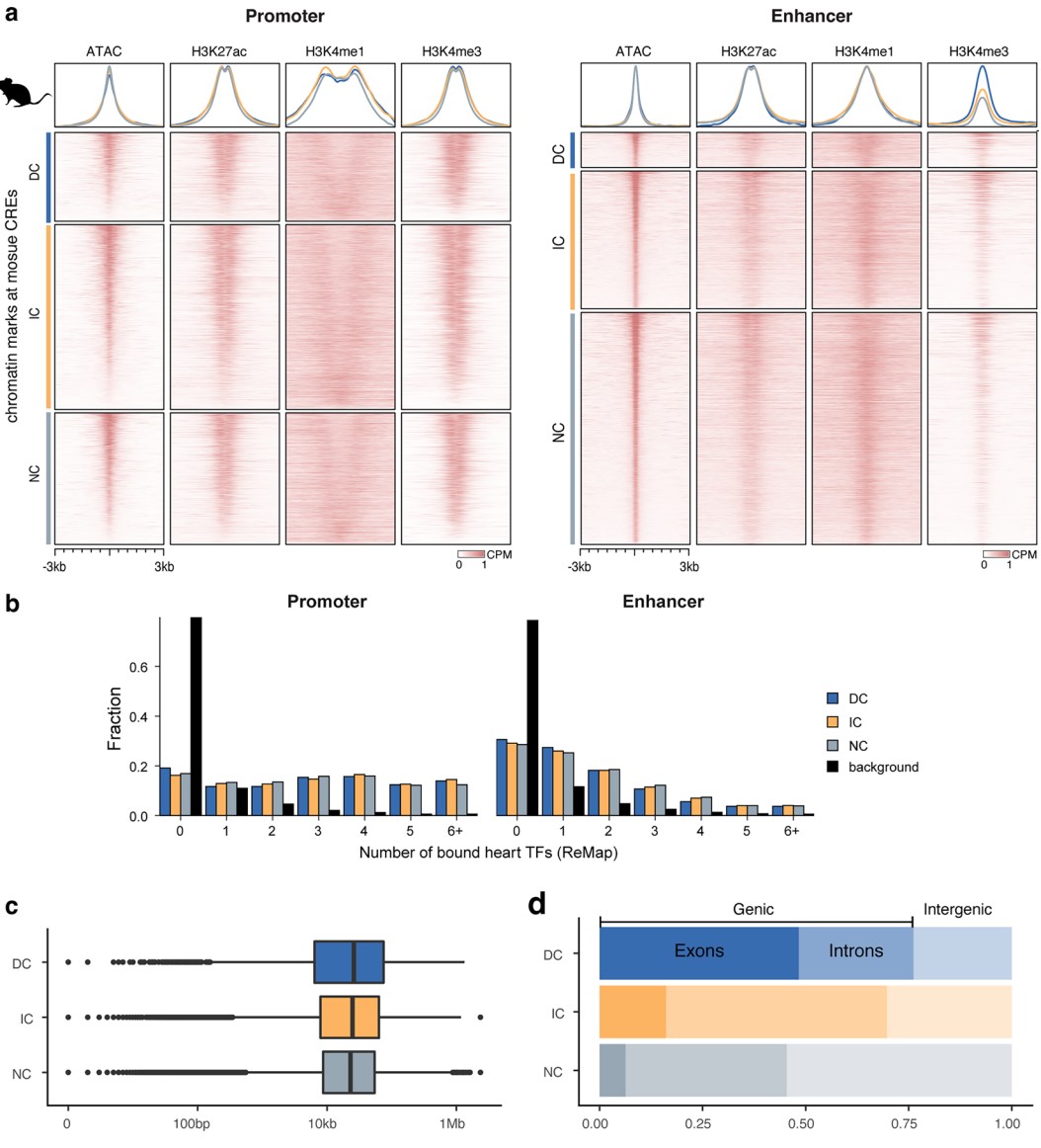

**Extended Data Fig. 4 | Functional and genomic characterization of DC/IC/NC enhancer classes in the mouse genome. (a)** Enrichment of promoter- and enhancer-specific histone modifications and ATAC-seq signal surrounding the distinct enhancer classes in mouse E10.5 heart samples. **(b)** Fraction of DC/IC/NC CREs overlapping one or multiple experimentally determined TFBSs (ReMap) vs. randomly selected genomic fragments. **(c)** Distance of enhancers to the nearest annotated TSS for DC/IC/NC enhancers **(d)** Fraction of enhancers located within annotated genes (and within genes overlapping exons or introns) as well as located in intergenic regions.

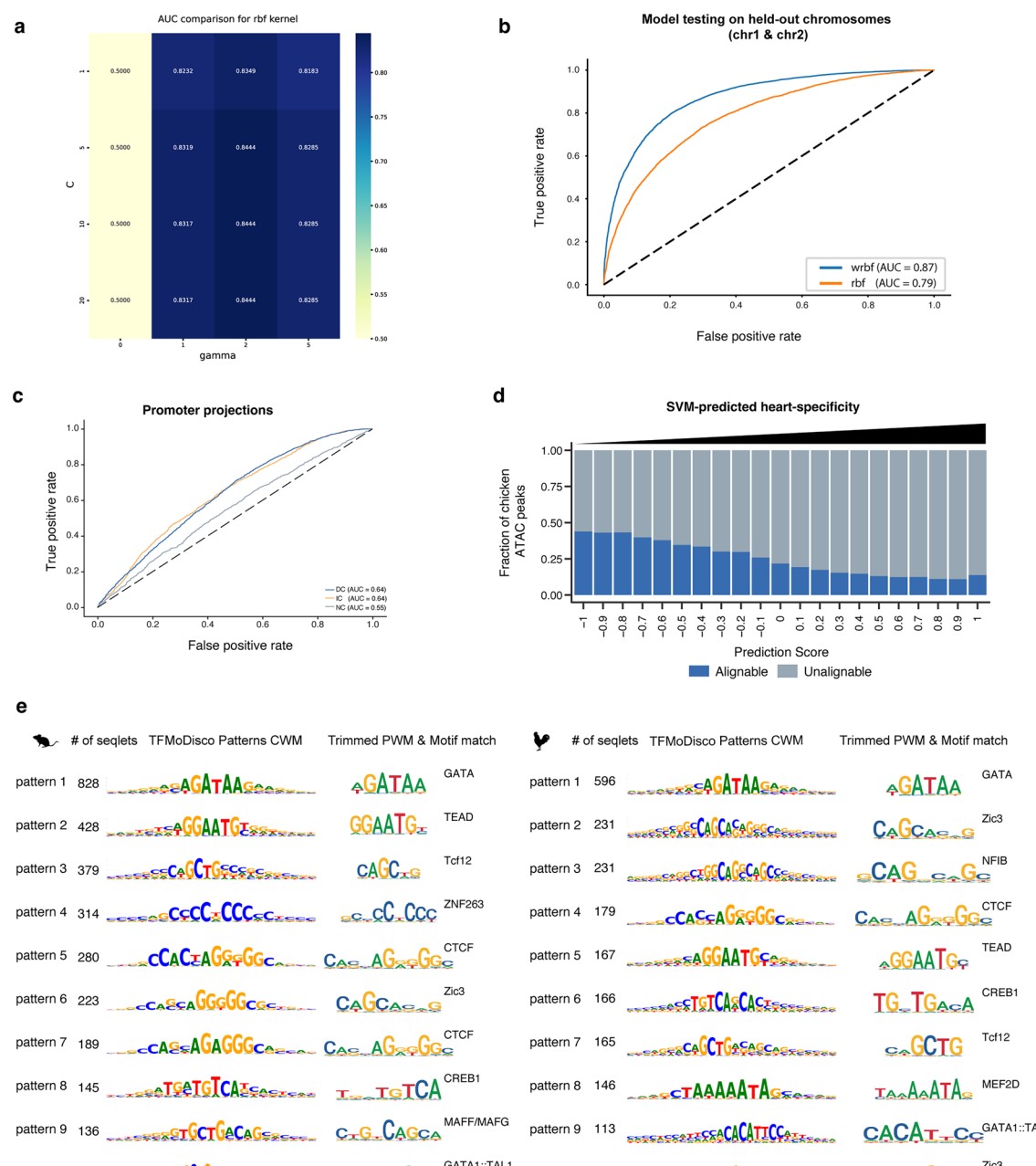

**Extended Data Fig. 5 | Validation of a machine learning model to predict heart specific enhancers across vertebrates. 5 (a)** Parameter tuning to train the SVM with RBF kernel with a grid-search for parameters c and gamma showing the calculated AUC after 5-fold cross validation. AUC = Area under the ROC curve. **(b)** ROC curves with computed AUC showing the performance of gkm-SVM with either RBF(rbf, orange) or weighted RBF(wrbf, blue) kernel on test data. The SVM was trained with the c & gamma parameters chosen in (a). **(c)** ROC curves with computed AUC showing human-chicken interspecies prediction accuracy for different conservation classes of mouse promoters projected to chicken. **(d)** Estimation of sequence alignability as a function of SVM predicted tissue-specificity (as prediction score) for ATAC-Seq peaks from chicken embryonic heart. **(e)** Top 10 mouse (left) and chicken (right) patterns discovered by TF-MoDisco showing seqlet as CWM, trimmed and converted PWMs and their annotated JASPAR motif match.

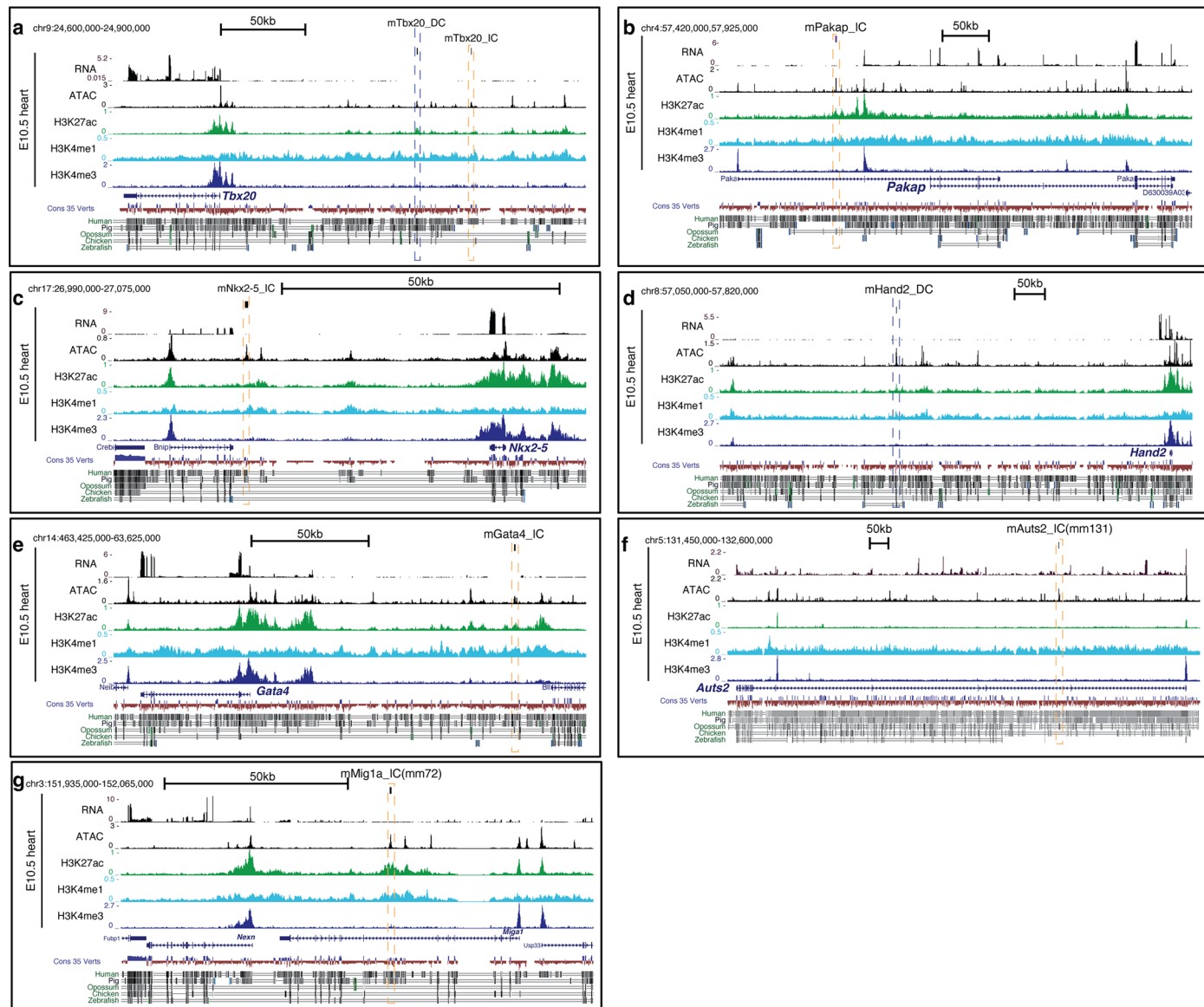

**Extended Data Fig. 6 | Genomic location of *in vivo* tested enhancers in the mouse genome. (a-g)** RNA-, ATAC-seq and ChIPmentation profiles from mouse E10.5 hearts show the distal location of tested IC/DC enhancers in the mouse genome. Scale bar: 50 kb.

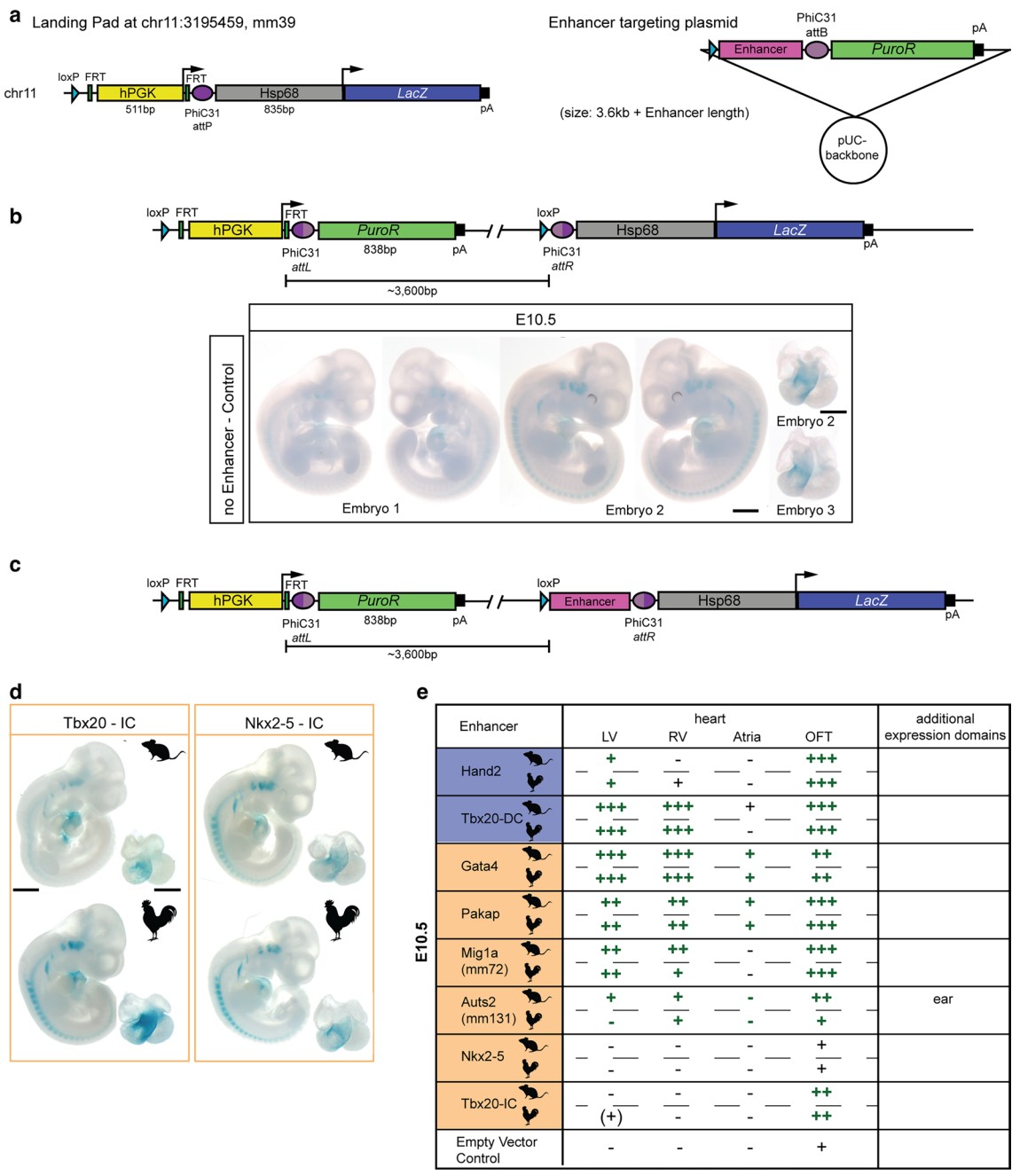

**Extended Data Fig. 7 | *In vivo* enhancer reporter assays test functional conservation of ortholog enhancer pairs. (a)** Landing pad at the H11 Locus and enhancer targeting plasmid **(b)** Control Experiment using integration of a non-enhancer plasmid integrated at the H11 locus shows weak background signal in the otic vesicle, somites and the outflow tract at E10.5. **(c)** Genomic organization of the integrated enhancer-reporter construct at the H11 locus **(d)** Enhancer-reporter results for Nkx2-5 and Tbx20-IC heart enhancer pairs at E10.5 and 2 adult liver enhancer pairs from mouse and opossum tested in E11.5 embryos. **(e)** Summary of enhancer activity from all enhancer-reporter assays tested in this study. Scale bar: 1,000 µm (embryo) or 500 µm (heart).

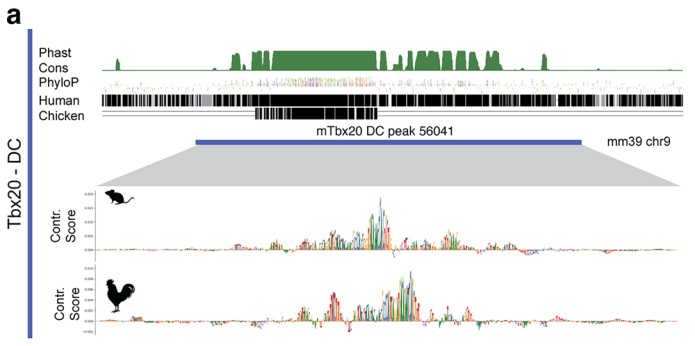

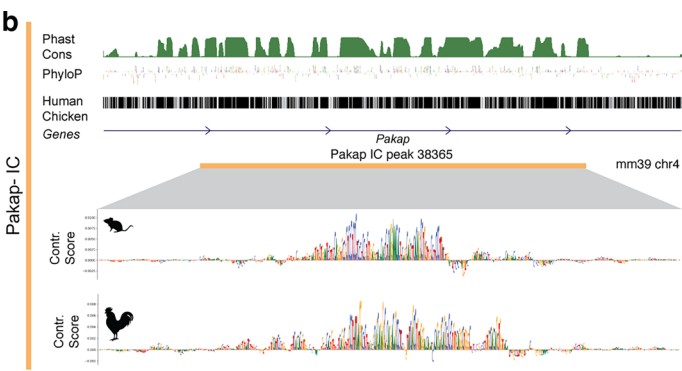

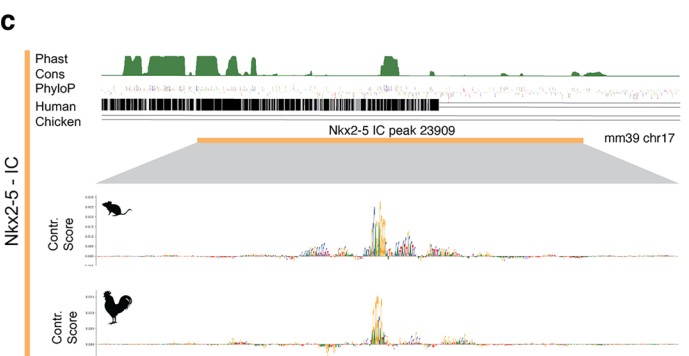

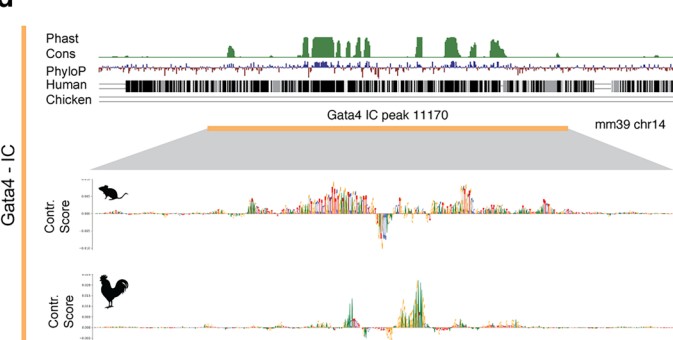

**Extended Data Fig. 8 | Comparison of ortholog *in vivo* tested enhancer pairs for their sequence conservation and transcription factor binding site composition. (a-d)** Sequence conservation scores (PhastCons/PhyloP) and direct alignments to human and chicken of tested enhancers at the Tbx20 (**a**), Pakap (**b**), Nkx2-5 (**c**) and Gata4 (**d**) loci. SVM contribution scores show important sequence features of the 500 bp enhancers in the mouse (top) and chicken (bottom) sequences tested.

# Reporting Summary

## Statistics

For all statistical analyses, confirm that the following items are present in the figure legend, table legend, main text, or Methods section.

| n/a | Confirmed | |
|---|---|---|
| ☐ | ☒ | The exact sample size (*n*) for each experimental group/condition, given as a discrete number and unit of measurement |
| ☒ | ☐ | A statement on whether measurements were taken from distinct samples or whether the same sample was measured repeatedly |
| ☐ | ☒ | The statistical test(s) used AND whether they are one- or two-sided<br>*Only common tests should be described solely by name; describe more complex techniques in the Methods section.* |
| ☒ | ☐ | A description of all covariates tested |
| ☒ | ☐ | A description of any assumptions or corrections, such as tests of normality and adjustment for multiple comparisons |
| ☐ | ☒ | A full description of the statistical parameters including central tendency (e.g. means) or other basic estimates (e.g. regression coefficient) AND variation (e.g. standard deviation) or associated estimates of uncertainty (e.g. confidence intervals) |
| ☐ | ☒ | For null hypothesis testing, the test statistic (e.g. *F*, *t*, *r*) with confidence intervals, effect sizes, degrees of freedom and *P* value noted<br>*Give P values as exact values whenever suitable.* |
| ☒ | ☐ | For Bayesian analysis, information on the choice of priors and Markov chain Monte Carlo settings |
| ☒ | ☐ | For hierarchical and complex designs, identification of the appropriate level for tests and full reporting of outcomes |
| ☐ | ☒ | Estimates of effect sizes (e.g. Cohen's *d*, Pearson's *r*), indicating how they were calculated |

*Our web collection on statistics for biologists contains articles on many of the points above.*

## Software and code

Policy information about availability of computer code

| Data collection | No software was used to collect data |
|---|---|
| Data analysis | Details about analysis are provided in the Methods section.<br>Code to run IPP can be found at https://github.com/tobiaszehnder/IPP<br><br>Softwares used to process sequencing libraries:<br>- samtools v1.10<br>- bowtie2 v.2.3.5<br>- deepTools<br>- BWA-mem v0.7.17<br>- cutadapt v 1.10<br>- Juicer v1.6.0<br>- STARv2.7.9a<br>- Genrich v0.6.1 https://github.com/jsh58/Genrich<br>- Picard v2.23.4<br>- FANC 0.9.25<br>- CRUP https://github.com/VerenaHeinrich/CRUP<br>- TOBIAS https://github.com/loosolab/TOBIAS<br><br>Main R packages used (for R version 4.2.1)<br>- tidyverse 1.3.2 |

- rtracklayer 1.58
- DESeq2 1.36
- universalmotif 1.14.1
- clusterProfiler 4.4.4
- memes 1.4.1
- BSgenome 1.66.3
- rankdist 1.1.4
- biomaRt 2.52.0
- gkmSVM 0.83

Softwares for building SVM model (Python 3.8.5)
- lsgkm https://github.com/kundajelab/lsgkm-svr
- GkmExplain https://github.com/kundajelab/lsgkm-svr
- TFMoDisco https://github.com/jmschrei/tfmodisco-lite

For manuscripts utilizing custom algorithms or software that are central to the research but not yet described in published literature, software must be made available to editors and reviewers. We strongly encourage code deposition in a community repository (e.g. GitHub). See the Nature Portfolio guidelines for submitting code & software for further information.

## Data

Policy information about availability of data

All manuscripts must include a data availability statement. This statement should provide the following information, where applicable:
- Accession codes, unique identifiers, or web links for publicly available datasets
- A description of any restrictions on data availability
- For clinical datasets or third party data, please ensure that the statement adheres to our policy

ChIPmentation, RNA-seq, ATAC-seq and Hi-C sequencing data generated from chicken embryonic heart and forelimb and mouse embryonic heart ATAC-seq data study have been deposited to NCBI GEO under GSE263587, GSE263753, GSE263755, GSE263783.
Following published dataset were re-analysed with details listed in Sup Table 2(GSM2544836, GSE185775, GSE185775, GSE185775, GSE185775, GSE185775, ENCSR582SPN, ENCSR266JQW, ENCSR782DGO, ENCSR782DEA, GSE185775, GSE185775, GSE185775, GSE185775, GSE185775, GSE185775, ENCSR222IHX, GSE185775, ENCSR963OLG, GSE185775, ENCSR886IHN, ENCSR592GQI, GSE164737, GSE164738)

## Research involving human participants, their data, or biological material

Policy information about studies with human participants or human data. See also policy information about sex, gender (identity/presentation), and sexual orientation and race, ethnicity and racism.

| Reporting on sex and gender | not applicable |
| Reporting on race, ethnicity, or other socially relevant groupings | not applicable |
| Population characteristics | not applicable |
| Recruitment | not applicable |
| Ethics oversight | not applicable |

Note that full information on the approval of the study protocol must also be provided in the manuscript.

# Field-specific reporting

Please select the one below that is the best fit for your research. If you are not sure, read the appropriate sections before making your selection.

☒ Life sciences   ☐ Behavioural & social sciences   ☐ Ecological, evolutionary & environmental sciences

For a reference copy of the document with all sections, see nature.com/documents/nr-reporting-summary-flat.pdf

# Life sciences study design

All studies must disclose on these points even when the disclosure is negative.

| Sample size | No prior analyses were used to determine the sample size of 2 biological replicates per ChIP/ATAC/RNA-seq and Hi-C experiment, but are based on standards in the field. Testing of 3 or more lacZ transgenic embryos from independent tetraploid aggregations was determined to generate sufficient embryos for lacZ stainings |

| Data exclusions | The embryos that were not at the correct developmental stage were excluded from data collection. |
| --- | --- |
| Replication | For lacZ enhancer-reporter assays, at least 3 animals/embryos of the appropriate genotype were stained and produced reproducible staining/phenotypes. Functional genomic experiments were produced in two biological replicates for each species and developmental stage |
| Randomization | There was no randomization of samples in this study |
| Blinding | Blinding was not relevant for our study, as enhancer-Reporter assays needed meticulous tracing of plasmids, cell cultures, and foster mothers to avoid mix-up. Consequently formal blinding of the experimental result was not possible. The resutls of the biological samples are not impacted by the unblinded design |

# Reporting for specific materials, systems and methods

We require information from authors about some types of materials, experimental systems and methods used in many studies. Here, indicate whether each material, system or method listed is relevant to your study. If you are not sure if a list item applies to your research, read the appropriate section before selecting a response.

## Materials & experimental systems

| n/a | Involved in the study |
| --- | --- |
| ☐ | ☒ Antibodies |
| ☐ | ☒ Eukaryotic cell lines |
| ☒ | ☐ Palaeontology and archaeology |
| ☐ | ☒ Animals and other organisms |
| ☒ | ☐ Clinical data |
| ☒ | ☐ Dual use research of concern |
| ☒ | ☐ Plants |

## Methods

| n/a | Involved in the study |
| --- | --- |
| ☐ | ☒ ChIP-seq |
| ☒ | ☐ Flow cytometry |
| ☒ | ☐ MRI-based neuroimaging |

## Antibodies

| Antibodies used | H3K4me1 (Diagenode #C15410037), H3K4me3 (Merck-Millipore #07-473), H3K27ac (Diagenode #C15410174) |
| --- | --- |
| Validation | Antibodies were validated in independent ChIP experiments on the manufacturer's website using ChIP-qPCR, ChIP-seq, Western Blots, Immunfluorescence and ELISAs. |

## Eukaryotic cell lines

Policy information about cell lines and Sex and Gender in Research

| Cell line source(s) | G4-ESCs were obtained from Anders Nagy and subsequently used to generate custom genome-engineered cell lines for generation of mice used in this study. |
| --- | --- |
| Authentication | The pluripotent state of the ESCs used was authenticated by generation of highly chimeric, germ-line transmitting mice through di- and tetraploid complementation assays |
| Mycoplasma contamination | all cell lines were tested negative for mycoplasma contamination |
| Commonly misidentified lines (See ICLAC register) | No commonly misidentified cell lines were used |

## Animals and other research organisms

Policy information about studies involving animals; ARRIVE guidelines recommended for reporting animal research, and Sex and Gender in Research

| Laboratory animals | Mouse lines described in this study were C57Bl.6/J mice genotype and chicken material was obtained as fertilized SPF eggs purchased from Valo Biomedia. |
| --- | --- |
| Wild animals | not applicable |
| Reporting on sex | Sex was not determined for embryo collection, but cohorts were presumed to include roughly equal numbers of males and females. |
| Field-collected samples | not applicable |
| Ethics oversight | The study plan was approved by the Landesamt für Gesundheit und Soziales (LaGeSo), Berlin under licenses G0243/18 and G0098/23. |

Note that full information on the approval of the study protocol must also be provided in the manuscript.

## Plants

Seed stocks

not applicable

Novel plant genotypes

not applicable

Authentication

not applicable

## ChIP-seq

### Data deposition

☒ Confirm that both raw and final processed data have been deposited in a public database such as GEO.

☒ Confirm that you have deposited or provided access to graph files (e.g. BED files) for the called peaks.

Data access links
*May remain private before publication.*

ChIPmentation, RNA-seq, ATAC-seq and Hi-C sequencing data generated from chicken embryonic heart and forelimb and mouse embryonic heart ATAC-seq data study have been deposited to NCBI GEO under GSE263587, GSE263753, GSE263755, GSE263783.

Files in database submission

H3K4me3_heart_HH24_galGal6_WT_Rep2_R2_001.fastq.gz
H3K4me3_heart_HH24_galGal6_WT_Rep2_R1_001.fastq.gz
H3K27ac_heart_HH22_galGal6_WT_Rep2_R1_001.fastq.gz
H3K4me1_heart_HH22_galGal6_WT_Rep1_R1_001.fastq.gz
H3K27ac_heart_HH22_galGal6_WT_Rep2_R2_001.fastq.gz
H3K4me1_heart_HH22_galGal6_WT_Rep1_R2_001.fastq.gz
input_heart_HH24_galGal6_WT_Rep1_R1_001.fastq.gz
H3K4me3_FL_HH22_galGal6_WT_Rep1_R1_001.fastq.gz
H3K4me3_FL_HH22_galGal6_WT_Rep1_R2_001.fastq.gz
H3K4me3_FL_HH24_galGal6_WT_Rep1_R2_001.fastq.gz
H3K27ac_heart_HH22_galGal6_WT_Rep1_R2_001.fastq.gz
H3K4me3_heart_HH22_galGal6_WT_Rep1_R2_001.fastq.gz
H3K4me3_heart_HH22_galGal6_WT_Rep1_R1_001.fastq.gz
H3K4me3_FL_HH24_galGal6_WT_Rep1_R1_001.fastq.gz
H3K4me1_heart_HH24_galGal6_WT_Rep2_R2_001.fastq.gz
H3K27ac_heart_HH24_galGal6_WT_Rep2_R2_001.fastq.gz
H3K27ac_heart_HH22_galGal6_WT_Rep1_R1_001.fastq.gz
H3K27ac_heart_HH24_galGal6_WT_Rep2_R1_001.fastq.gz
H3K27ac_FL_HH24_galGal6_WT_Rep1_R2_001.fastq.gz
H3K27ac_heart_HH24_galGal6_WT_Rep1_R2_001.fastq.gz
H3K27ac_heart_HH24_galGal6_WT_Rep1_R1_001.fastq.gz
H3K4me1_heart_HH24_galGal6_WT_Rep2_R1_001.fastq.gz
H3K27ac_FL_HH24_galGal6_WT_Rep1_R1_001.fastq.gz
H3K4me1_heart_HH22_galGal6_WT_Rep2_R1_001.fastq.gz
H3K4me1_heart_HH22_galGal6_WT_Rep2_R2_001.fastq.gz
H3K4me1_FL_HH22_galGal6_WT_Rep1_R2_001.fastq.gz
H3K27ac_FL_HH22_galGal6_WT_Rep1_R2_001.fastq.gz
H3K27ac_FL_HH22_galGal6_WT_Rep1_R1_001.fastq.gz
H3K4me1_FL_HH22_galGal6_WT_Rep1_R1_001.fastq.gz
H3K4me3_heart_HH22_galGal6_WT_Rep2_R1_001.fastq.gz
H3K4me3_heart_HH22_galGal6_WT_Rep2_R2_001.fastq.gz
H3K4me1_FL_HH24_galGal6_WT_Rep1_R2_001.fastq.gz
H3K4me1_FL_HH24_galGal6_WT_Rep1_R1_001.fastq.gz
H3K4me1_heart_HH24_galGal6_WT_Rep1_R1_001.fastq.gz
H3K4me1_heart_HH24_galGal6_WT_Rep1_R2_001.fastq.gz
H3K4me3_heart_HH24_galGal6_WT_Rep1_R1_001.fastq.gz
H3K4me3_heart_HH24_galGal6_WT_Rep1_R2_001.fastq.gz
H3K4me1_FL_HH24_galGal6_WT_Rep2_R2_001.fastq.gz
H3K27ac_FL_HH24_galGal6_WT_Rep2_R1_001.fastq.gz
H3K4me1_FL_HH24_galGal6_WT_Rep2_R1_001.fastq.gz
H3K27ac_FL_HH24_galGal6_WT_Rep2_R2_001.fastq.gz
H3K4me3_FL_HH24_galGal6_WT_Rep2_R2_001.fastq.gz
H3K4me3_FL_HH24_galGal6_WT_Rep2_R1_001.fastq.gz
H3K4me3_FL_HH24_galGal6_WT_Rep2.cpm.bw
H3K27ac_FL_HH24_galGal6_WT_Rep2.cpm.bw

H3K27ac_heart_HH24_galGal6_WT_Rep1.cpm.bw
H3K4me3_heart_HH24_galGal6_WT_Rep2.cpm.bw
H3K4me1_FL_HH24_galGal6_WT_Rep2.cpm.bw
H3K4me3_heart_HH22_galGal6_WT_Rep2.cpm.bw
H3K4me1_heart_HH24_galGal6_WT_Rep2.cpm.bw
H3K4me1_heart_HH22_galGal6_WT_Rep2.cpm.bw
H3K4me3_FL_HH24_galGal6_WT_Rep1.cpm.bw
H3K4me3_heart_HH24_galGal6_WT_Rep1.cpm.bw
H3K4me1_FL_HH24_galGal6_WT_Rep1.cpm.bw
H3K4me3_FL_HH22_galGal6_WT_Rep1.cpm.bw
H3K4me3_heart_HH22_galGal6_WT_Rep1.cpm.bw
H3K27ac_FL_HH24_galGal6_WT_Rep1.cpm.bw
H3K27ac_heart_HH24_galGal6_WT_Rep2.cpm.bw
H3K27ac_heart_HH22_galGal6_WT_Rep2.cpm.bw
H3K4me1_heart_HH22_galGal6_WT_Rep1.cpm.bw
H3K4me1_heart_HH24_galGal6_WT_Rep1.cpm.bw
H3K27ac_heart_HH22_galGal6_WT_Rep1.cpm.bw
H3K27ac_FL_HH22_galGal6_WT_Rep1.cpm.bw
H3K4me1_FL_HH22_galGal6_WT_Rep1.cpm.bw
ATAC-seq_FL_HH22_galGal6_WT_Rep2_R1_001.fastq.gz
ATAC-seq_FL_HH22_galGal6_WT_Rep2_R2_001.fastq.gz
ATAC-seq_heart_HH24_galGal6_WT_Rep1_R2_001.fastq.gz
ATAC-seq_heart_HH24_galGal6_WT_Rep2_R2_001.fastq.gz
ATAC-seq_heart_HH24_galGal6_WT_Rep1_R1_001.fastq.gz
ATAC-seq_heart_HH24_galGal6_WT_Rep2_R1_001.fastq.gz
ATAC-seq_FL_HH24_galGal6_WT_Rep1_R2_001.fastq.gz
ATAC-seq_FL_HH24_galGal6_WT_Rep1_R1_001.fastq.gz
ATAC-seq_FL_HH24_galGal6_WT_Rep2_R2_001.fastq.gz
ATAC-seq_FL_HH24_galGal6_WT_Rep2_R1_001.fastq.gz
ATAC-seq_heart_E115_mm39_WT_Rep1_R2_001.fastq.gz
ATAC-seq_heart_E115_mm39_WT_Rep1_R1_001.fastq.gz
ATAC-seq_heart_E115_mm39_WT_Rep2_R2_001.fastq.gz
ATAC-seq_heart_E115_mm39_WT_Rep2_R1_001.fastq.gz
ATAC-seq_heart_E105_mm39_WT_Rep2_R1_001.fastq.gz
ATAC-seq_heart_E105_mm39_WT_Rep2_R2_001.fastq.gz
ATAC-seq_heart_E105_mm39_WT_Rep1_R2_001.fastq.gz
ATAC-seq_heart_E105_mm39_WT_Rep1_R1_001.fastq.gz
ATAC-seq_heart_HH22_galGal6_WT_Rep2_R2_001.fastq.gz
ATAC-seq_heart_HH22_galGal6_WT_Rep2_R1_001.fastq.gz
ATAC-seq_heart_HH22_galGal6_WT_Rep1_R1_001.fastq.gz
ATAC-seq_heart_HH22_galGal6_WT_Rep1_R2_001.fastq.gz
ATAC-seq_FL_HH22_galGal6_WT_Rep1_R2_001.fastq.gz
ATAC-seq_FL_HH22_galGal6_WT_Rep1_R1_001.fastq.gz
ATAC-seq_FL_HH22_galGal6_WT_Rep2.cpm.bw
ATAC-seq_heart_HH22_galGal6_WT_Rep2.cpm.bw
ATAC-seq_heart_HH22_galGal6_WT_Rep1.cpm.bw
ATAC-seq_FL_HH24_galGal6_WT_Rep2.cpm.bw
ATAC-seq_FL_HH24_galGal6_WT_Rep1.cpm.bw
ATAC-seq_heart_HH24_galGal6_WT_Rep1.cpm.bw
ATAC-seq_FL_HH22_galGal6_WT_Rep1.cpm.bw
ATAC-seq_heart_E105_mm39_WT_Rep2.cpm.bw
ATAC-seq_heart_HH24_galGal6_WT_Rep2.cpm.bw
ATAC-seq_heart_E115_mm39_WT_Rep1.cpm.bw
ATAC-seq_heart_E115_mm39_WT_Rep2.cpm.bw
ATAC-seq_heart_E105_mm39_WT_Rep1.cpm.bw
HiC_heart_HH22_galGal6_WT_Rep1_R1_001.fastq.gz
HiC_heart_HH22_galGal6_WT_Rep2_R1_001.fastq.gz
HiC_heart_HH22_galGal6_WT_Rep1_R2_001.fastq.gz
HiC_heart_HH22_galGal6_WT_Rep2_R2_001.fastq.gz
RNA-seq_heart_HH22_galGal6_WT_Rep1_R1_001.fastq.gz
RNA-seq_heart_HH22_galGal6_WT_Rep1_R2_001.fastq.gz
RNA-seq_heart_HH22_galGal6_WT_Rep2_R1_001.fastq.gz
RNA-seq_heart_HH22_galGal6_WT_Rep2_R2_001.fastq.gz
RNA-seq_heart_HH24_galGal6_WT_Rep1_R1_001.fastq.gz
RNA-seq_heart_HH24_galGal6_WT_Rep1_R2_001.fastq.gz
RNA-seq_heart_HH24_galGal6_WT_Rep2_R1_001.fastq.gz
RNA-seq_heart_HH24_galGal6_WT_Rep2_R2_001.fastq.gz

Genome browser session
(e.g. UCSC)

https://genome-euro.ucsc.edu/s/mphan236/heart_chromatin_mm39
https://genome-euro.ucsc.edu/s/mphan236/heart_chromatin_galGal6

## Methodology

Replicates

all ChIP, ATAC and RNA-seq experiments were performed in two biological replicates per sample, species, and developmental stage

| | |
|---|---|
| Sequencing depth | All libraries were sequenced with 100bp pair-end. Sequencing depth for each biological replicate was 100 million fragments for ATAC-seq and 30-50 million fragments for ChIPmentation. |
| Antibodies | H3K4me1 (Diagenode #C15410037), H3K4me3 (Merck-Millipore #07-473), H3K27ac (Diagenode #C15410174) |
| Peak calling parameters | ATAC peak calling was done using Genrich v0.6.1 using '-j' mode and default parameters |
| Data quality | Aligned reads with MAPQ <10 were excluded. Duplicated, unmapped, and unpaired reads were also filtered out. |
| Software | - samtools v1.10<br>- bowtie2 v.2.3.5<br>- deepTools<br>- BWA-mem v0.7.17<br>- cutadapt v 1.10<br>- Genrich v0.6.1 https://github.com/jsh58/Genrich<br>- Picard v2.23.4 |

