## [Peer Review File · Nature Genetics]

Conservation of Regulatory Elements with Highly Diverged Sequences Across Large Evolutionary Distances

Corresponding Author: Dr Daniel Ibrahim

Version 0:

Decision Letter:

17th Jun 2024

Dear Daniel,

Your Article, "Conservation of Regulatory Elements with Highly Diverged Sequences Across Large Evolutionary Distances" has now been seen by 3 referees. You will see from their comments copied below that while they find your work of considerable potential interest, they have raised quite substantial concerns that must be addressed. In light of these comments, we cannot accept the manuscript for publication, but would be very interested in considering a revised version that addresses these serious concerns.

We hope you will find the referees' comments useful as you decide how to proceed. If you wish to submit a substantially revised manuscript, please bear in mind that we will be reluctant to approach the referees again in the absence of major revisions.

To guide the scope of the revisions, the editors discuss the referee reports in detail within the team, including with the chief editor, with a view to identifying key priorities that should be addressed in revision and sometimes overruling referee requests that are deemed beyond the scope of the current study. In this case, we would like you to address Reviewers' comments in full. In particular, we invite you to further prove the advantage of your method by experimentally testing other less-obvious candidates, in line with Reviewer #2's request. We hope that you will find the prioritised set of referee points to be useful when revising your study. Please do not hesitate to get in touch if you would like to discuss these issues further.

If you choose to revise your manuscript taking into account all reviewer and editor comments, please highlight all changes in the manuscript text file. At this stage we will need you to upload a copy of the manuscript in MS Word .docx or similar editable format.

*2) If you have not done so already please begin to revise your manuscript so that it conforms to our Article format instructions, available [here](http://www.nature.com/ng/authors/article_types/index.html). Refer also to any guidelines provided in this letter.

*3) Include a revised version of any required Reporting Summary: <https://www.nature.com/documents/nr-reporting-summary.pdf> It will be available to referees (and, potentially, statisticians) to aid in their evaluation if the manuscript goes back for peer review.

Link Redacted

If you wish to submit a suitably revised manuscript we would hope to receive it within 6 months. If you cannot send it within this time, please let us know. We will be happy to consider your revision so long as nothing similar has been accepted for publication at Nature Genetics or published elsewhere. Should your manuscript be substantially delayed without notifying us in advance and your article is eventually published, the received date would be that of the revised, not the original, version.

Thank you for the opportunity to review your work and sorry for the delay in providing our decision.

All the best,
Chiara

Chiara Anania, PhD
Associate Editor
Nature Genetics
<https://orcid.org/0000-0003-1549-4157>

Referee expertise:

Referee #1: sequence-encoded function of enhancers; method development

Referee #2: gene regulation

Referee #3: evo; comparative epigenomics

Reviewers' Comments:

Reviewer #1:

Remarks to the Author:

Phan and co-authors use a synteny and inter-species alignment bridging approach to identify and compare the features of enhancers with different sequence conservation profiles in mouse and chicken embryonic heart development. First, the authors generate chromatin and gene expression profiles (histone ChIP-seq, ATAC-seq, HiC, RNA-seq) from a matched stage of development between mouse and chicken. Following annotation of promoters and enhancers, the authors then ask how much of the regulatory landscape is conserved between the evolutionarily divergent species. Similar to previous studies, they found that many of the regulatory element sequences were not easily aligned, thus not strongly conserved at the sequence level. The main novelty of the study comes from the strategy they used to map sequences by synteny and indirect mapping through the use of secondary "bridge" species. They use a method they previously developed, Interspecies Point Projection (Baranasic et al. 2022), which projects synteny/conservation via proximity to alignable relationships across other genomes, and relationships identified via this analysis are defined as "indirect conservation" or IC, compared to direct conservation (DC). They then compare features of DC, IC and non-conserved (NC) enhancers, finding reasonably similar functional activity and motif composition between DC and IC loci. IC enhancers had increased motif shuffling, but similar levels of motif predictive utility in a machine learning model. Overall, this is a nice work that furthers understanding of the evolutionary history of enhancers and the relationship between sequence and function. While the methods are robust and rigorously deployed, the study does not adequately explore the evolutionary relationships of IC enhancers or test to see if this model holds at later developmental stages (or other tissues), when functional conservation has been described to be minimal. Overall, I think this is a strong study with a lot to offer to the community and that my criticisms should be addressable without significant new experiments/cost.

1. The authors very briefly describe what is required for DC status via UCSC LiftOver. While this is a fine choice, as liftOver is widely used and enables simple yes/no calling, this is a fairly simple way to look at sequence conservation. It would be of interest to test how relaxing alignment parameters (e.g using smaller intervals or changing from minmatch=.1) or using

different alignment approaches changes how direct sequence conservation to be captured. Related to the above point, it would be of interest to do a deeper dive on the homology that is present for the IC class. It seems that the synteny model would be that sequences that hit IC thresholds should be largely descended from common ancestral sequence? Or could these be insertions of sequences that are not presumed to be evolutionarily shared? As an example, do paired IC/NC sequences map using other sequence homology methods that might permit reduced alignment (e.g. BLAST), or are they so divergent as to be unalignable or stemming from retroelement insertions? Without deeper investigation, the methods advancement is somewhat murky as the IPP methods are already published and the biological relevance is muted as the new insights are limited with regard to the origins and make-up of IC enhancers.

2. More analysis should be done on comparing the three enhancer classes (DC/IC/NC) – the current paragraph is not very informative. The present manuscript states similar chromatin state (H3K4me1 at enhancers and H3K4me3 at promoters) and overall similar chromosomal contacts. But the authors should have the data to dive deeper. Is there a difference in HiC/contacts or TAD positioning for DC/IC/NC loci? Is there a difference in presence of H3K27ac or in strength of histone ChIP-seq signal? What is the distribution of DC/IC/NC enhancers with regards to an average genomic interval? For example, are DC/IC more or less interchangeable in the average number per gene/locus? Are NC enhancers more likely to be found near DC/IC enhancers?

3. It would be of interest to know how the DC/IC/NC proportions for annotated enhancers varies across tissues/developmental stages. As the authors point out, once the projections are done, it is easy to extend to other regulatory loci sets (as done for liver TF ChIP and limb enhancers in the manuscript). It would be of interest to compare more broadly, e.g. to some of the mouse vs. human tissue samples in ENCODE. This would increase the impact and enable some overarching comparisons about enhancer conservation.

4. Minor point – it is unclear how the two timepoints are used. In some places, both timepoints are stated (E10.5/E11.5) compared to others where only one timepoint. The supp fig shows some presumed stage-specific differences. How do DC/IC/NC stats compare for the timepoint specific vs. shared loci? The manuscript should clarify when and how the data is used for the multiple timepoints.

Reviewer #2:

Remarks to the Author:

The manuscript by Phan et al addresses an important challenge, how to identify orthologous enhancers between distantly related species, as many cis regulatory elements cannot be aligned. The authors integrate two approaches – synteny and the use of intermediate species – which have both been used independently to identify orthologous regions. Here the novelty is to combine both, which negates the need to rely on sequence alignment.

I was very excited by the idea when I read the abstract, but my enthusiasm is a bit tempered after reading the manuscript. The authors compare their method to other alignment-based methods using hierarchical trees, and conclude that their method performs as similarly or better at relatively short evolutionary distances, but has a clear advantage over large distances (Lines 142), as it identifies more putative orthologous enhancers. The key question then is are the regions that they discover in distally related species really new enhancers? Unfortunately, the data provided falls short of proving that. The authors use 3 metrics (discussed below) to show that their newly identified enhancers with indirect conservation (IC enhancers) are real, with similar properties as enhancers with direct conservation (DC): In vivo testing, chromatin signatures and motif enrichment. All three methods showed some evidence for success, especially the two global ones, each has issues as discussed below.

1) In vivo testing in transgenic embryos – which is the definitive proof.

It's great that the authors tested some chick IC enhancers in mouse –

But unfortunately, the four IC regions tested were biased – selected around the four major heart lineage genes, Hand2, Tbx20, Nkx2.5, Gata4 – these are the main conserved lineage factors in all species for the heart. If one just took chick ATAC peaks with H3K27ac signal in the vicinity of these genes, there is a very high probability that they would also give heart activity in mouse. The authors didn't show that the IPP gave any advantage here.

Also, how exactly were the tested regions selected, they just state “we selected two pairs of DC and 4 pairs of IC enhancers.” – no information is given in the text or methods. What distance to the genes was used? Did you use Phascons or other signatures of conservation or selection? Is there a confidence score from the IPP model? If yes, was this used?

To really show the advantage of the method – take some regions from poorly studied genes,

Or regions very distal that are not obvious where you could only find them with the IPP method and not by sequence alignment and/or chromatin signatures.

This would greatly strengthen the manuscript.

2) A more global assessment of IC vs DC enhancers with chromatin signatures:

Line 163 – As only the coordinates of the regions experimentally identified (using H3K27ac, ATAC data) in mouse were used by IPP to identify the regions in chicken, the authors could use the chicken chromatin data as validations – if the IPP IC regions are correct, they should have genomic signatures of enhancers and promoters, which all makes logical sense.

The results were good on the one hand, as they were comparable between DC and IC enhancers

For DC: 66% promoters and 29% enhancers had an ATAC peak in chick

For IC: 56% promoters, and 26% enhancers (line 170)

This is in line with each other – but even the percentages for the direct conserved regions are surprisingly low for enhancers – 30% are bound (ATAC peak). How does this compare to other methods, e.g. the hierarchical method mentioned?

3) A more global assessment of IC vs DC enhancers by motif analysis.

Using gkm-SVM – IC enhancers were equally likely to be classified as heart enhancers as DC enhancers (AUC ~0.75 for each), providing good evidence that there is a global signature there.

Although I don't agree with this statement

“known motifs of master regulators of heart development (e.g. GATA, TEAD and HAND) were most predictive of tissue specificity (Fig. 3c)” – line 211,

At least the data is not shown in Fig 3c – only the PWMs for the motifs.

Show a ranked ordered heatmap of the p-values of motifs, so we can see which are the most predictive.

Only the functionally conserved CREs (ie. with chromatin marks) have the same motifs for DC and IC regions.

However, IC+ enhancers shared as many motifs as with DC- regions - Line 225.

How do you explain that? It does not fit with a 'shared syntax to DC+ enhancers.

Figure 3f (please note the text refers to Fig 4f) has a very weak signature for IC enhancers

There is little/no evidence for a conserved motif syntax etc for the IC enhancers, as it states in the text (line 235 etc)

While I appreciate that the two global assessments discussed above (based on chromatin and motifs) of the IC enhancers gave some signatures, which shows that there is a signature there. However, neither of these analyses were done with other methods (e.g. hallifover/HALPER). Which enhancers did this method identify that IPP didn't? Do those enhancers also have chromatin and motif signatures suggestive of heart enhancers? In other words how well is the IPP method performing to identify real functional enhancers compared to other methods?

4) The authors provide the source code in Github, which is great. But as this manuscript is primarily a new computational method, to be generally useful and a significant advance, the authors should provide the IPP method as a user-friendly package (in BioConductor or elsewhere).

Minor comments

1) How was the equivalent developmental stages defined between chicken and mouse?

This is central

2) What was the definition used for sequence conservation?

For example, at line 94/95, and line 130 – (Fig 2C)

22% promoters, 10% enhancers are conserved – the definition used to define something as conserved should be stated in the main text – this is crucial

3) What is the global conservation of TADs? Fig S1

4) Line 185 – give the reference for the gapped-kmer model

Line 186 “we first trained a gapped k-mer Support Vector Machine (gkm-SVM) model on mouse data to identify heart-specific enhancers.”

The model is not identifying heart enhancers from the rest of the genome de novo - you are providing it with the heart enhancers. It is rather learning the k-mers within heart enhancers – this sentence should be changed or merged with the next one

5) Fig 3c, it is surprising that Nkx2.5 motif is not present.

Alternatively, why is the CTCF motif enriched? – it should have been cancelled out in the background. How was the background matched for the gkm-SVM predictions in heart enhancers? Was another set of enhancers used? Or all other distal ATAC peaks?

Reviewer #3:

Remarks to the Author:

In this manuscript, Phan and colleagues describe a new strategy to assess enhancer conservation across species. Instead of relying solely on multiple sequence alignments, they use a combined approach that takes into account the synteny of neighbouring “anchor” regions and “bridging” species. This approach, termed IPP, identifies many enhancers positioned in syntenically orthologous regions across mouse and chicken, despite having divergent sequences. The authors argue that most of these sequences would be missed if using only sequence alignment scores, thus increasing the number of “conserved” enhancers across species significantly. They then validate that these positionally conserved enhancers are effective predictors of regulatory information using a combination of machine learning and experiments, demonstrating that enhancers from chicken can produce heart-specific expression patterns in mice. They find that these enhancers usually contain a similar number of transcription factor binding sites (TFBS), but in a re-arranged order, distinguishing them from enhancers that are conserved at the sequence level.

I find the manuscript well-written, clear in its analysis, and sound in its conclusions. However, the manuscript and the analysis tool presented are largely confirmatory of previous findings. As the authors note in the discussion, many examples of positionally conserved enhancers with shuffled TFBS have been reported, including some representing evolutionary

distances much larger than those reported in this manuscript (e.g. sponges PMID: 33154111 or hemichordates PMID: 27064252). The IPP approach supports the expected model of enhancer evolution, where rapid sequence turnover can produce functionally equivalent enhancers without requiring sequence conservation just by TFBS composition. Furthermore, the IPP method shows a similar rate of capturing these positionally conserved enhancers as the previous published HALPER pipeline that relied on multiple genome alignments. Thus, the most novel aspect of this work is the quantitative reappraisal of the number of conserved enhancers across species. While this might seem trivial, the landmark paper by Villar et al. (2015) concluded that most enhancers were not conserved even within mammals. This study reaffirms that positionally conserved enhancers, though not highly conserved at the sequence level, likely result from vertical evolution with frequent TFBS changes compromising the alignment. Having a clear view on how many enhancers are actually conserved at the level of sequence, how many are positionally conserved and functionally equivalent, and how many are actually new in a given species will have an impact on the field, as probably these behave differently due to distinct functional constraints.

The main points I think this manuscript should address are these:

1- The pipeline for IPP is deposited in GitHub, and that's great, but I have been looking at the README file and it does not look very well explained. In contrast, the github repository of HALPER is very detailed. Since the IPP approach would be a main sell of this work, as many groups might want to try similar things with their own set of genomes, I think the pipeline should be much more detailed, with a step-by-step explanation on how to obtain the input files, how to process them, how to run this approach, and what the output files look like. "python project.py" is the only command that is specified in the repository, which kind of speaks for itself. The authors cite the DANIO-CODE paper (Baranasic et al 2022) as a reference, but there I couldn't find a tool description more detailed than in the github associated with this repository. If the main advantage of this method is that it requires less input sequences than HALPER and is computationally less intensive, a widely usable code would increase the method's adoption by the community and the impact of this work.

2- Using the chicken versus mice comparison as the centre for the analysis is well justified, as heart enhancers have been particularly difficult to find using sequence conservation alone. Still, I think using an orthogonal dataset to some extent would also increase the potential impact of this work. For example, reanalysing Villar et al 2015 data, and highlighting the new numbers of conservation using the IPP pipeline would be impactful, as it might somehow overturn the quantitative expectations of the field (most enhancers are not conserved at the level of sequence, yet most are positionally).

3- Perhaps the most shocking result that is not really discussed are the DC- enhancers. These are sequence conserved, but not equivalently active in the chicken-mouse comparison. When analysing the TFBS composition, the IC+ (positionally conserved active in heart in both species) show a similar level of conservation of TFBS regulatory logic to DC-. Perhaps these DC- enhancers are pleiotropic and that's why they are conserved at the level of sequence but have suffered some change that makes them not active anymore in the heart of either species. But since they have an equivalent composition of TFBS, then what changed? Perhaps looking at silencing histone marks (probably already available for mice at least) could explain this divergence in function despite sequence conservation. Also, it would be important to see what patterns of expression some DC- enhancers from chicken drive in mice. Despite the higher sequence conservation, are they no longer capable of driving heart expression patterns? Or is a change in trans unique to chickens that silences the enhancers despite harbouring a regulatory lexicon capable of heart expression?

4 - Then, the authors should also do a motif analysis on the NC+ set of enhancers. Are these likely "novel" enhancers showing a different regulatory logic than the DC/IC? The TFBS conservation seems lower, but are the same motifs that dictate the general signal for the tissue also present there as the most enriched? Or the non-conserved enhancers are really the place to look for regulatory innovation? E.g. It would be interesting if the NC+ enhancers formed by using a new set of motifs not present in the "core" set of conserved enhancers. These could be transcription factors that are unique to one of the species (TF not expressed in either mouse or chicken), and therefore, the place to look for lineage specific-adaptations. The other possibility is that the novel enhancers might just reproduce the same regulatory logic as the conserved (DC or IC), which would then imply a neutral mode of evolution, where sequence turnover is more likely to act on some positions, but still can now and then produce enhancers from new genomic regions but largely using a conserved TF lexicon.

Then, I have some minor comments:

Line 218. "If IPP projections represent conserved pairs of CREs, these regions should share the same TFBS." Well, not really, conserved CREs might also have some differences on TFBS, not all should be conserved (some for sure). Perhaps rewriting as it "should share more TFBS in common than non-conserved enhancers".

Line 231 – Why is the background restricted to TAD? Motifs can be anywhere. Perhaps adding a minimal explanation in the main text instead of referring to methods would be a good clarification.

Version 1:

Decision Letter:

Our ref: NG-A65043R

29th Jan 2025

Dear Daniel,

thank you for submitting your revised manuscript "Conservation of Regulatory Elements with Highly Diverged Sequences Across Large Evolutionary Distances" (NG-A65043R). It has now been seen by the original referees and their comments are below. The reviewers find that the paper has improved in revision, and therefore we'll be happy in principle to publish it in Nature Genetics, pending minor revisions to satisfy the referees' final requests (please address the remaining suggestions by Reviewer #3) and to comply with our editorial and formatting guidelines.

Thank you again for your interest in Nature Genetics. Please do not hesitate to contact me if you have any questions.

Congratulations!

Best,
Chiara

Chiara Anania, PhD
Associate Editor
Nature Genetics
<https://orcid.org/0000-0003-1549-4157>

Reviewer #1 (Remarks to the Author):

The authors addressed my points and seem to have also covered major issues raised by other reviewers. The manuscript is greatly improved as is the accessibility/documentation of the software. I think the IPP is a nice addition to other methods to identify orthologous but not conserved enhancers and this manuscript will be of broad interest.

Reviewer #1 (Remarks on code availability):

I briefly reviewed the github page but did not attempt to implement myself.

Reviewer #2 (Remarks to the Author):

I appreciate that the authors added in two more IC enhancer pairs that were tested in vivo. I also appreciate that transgenic reporter assays are laborious and take a long time. I wasn't asking for many more to be tested, but rather for IC regions to be tested that you have no prior information on – regions not in the vicinity of a gene expressed in the heart (as the new Pakap-IC tested region is), or an enhancer known to give activity in the heart (like Mig1a and Auts2, which are shorter regions within known mouse heart enhancers). It would have been so much more impressive to take regions based purely on the IPP score, without looking if they are in the same TAD or close to a gene or enhancer with known expression in the heart in mouse. Such prerequisites in the selection of tested regions will of course increase the chances of success. Now 4 out of 6 IC enhancer pairs worked – only time will tell how many IC enhancer pairs would work if you are agnostic to the surrounding genes' expression, or presence within the same TAD.

In any case, I do appreciate the work. The authors have addressed all of my other comments very well, including the GitHub, and I therefore feel the manuscript should be published such that the community can assess the tool for themselves.

Reviewer #2 (Remarks on code availability):

I reviewed how well the Github is documented, but I didn't run the code

Reviewer #3 (Remarks to the Author):

I congratulate the authors for the revision. All my main points have been addressed, and frankly I think the resulting manuscript is stronger than its original form (which was already very good). I have also looked at the github repository and that's clearly an improvement. Confirming that a 3rd party user tried this tool successfully is a valuable point.

I have one suggestion (see code section) of including an option for custom "fasta" genome files as input for the pipeline, as that could make this tool more useful (e.g. non-UCSC genomes, custom assemblies).

Then, I just wanted to clarify that when I asked "4 - Then, the authors should also do a motif analysis on the NC+ set of enhancers. Are these likely "novel" enhancers showing a different regulatory logic than the DC/IC? The TFBS conservation

seems lower, but are the same motifs that dictate the general signal for the tissue also present there as the most enriched? Or the non-conserved enhancers are really the place to look for regulatory innovation?"

What I wanted to know, beyond what the overall conservation that figure 4e/f shows, is what kind of TFs bind to active non-conserved enhancers (NC+). NC+ are likely to be truly new (de novo) enhancers. So new enhancers could arise in two ways 1) gaining by chance a "core" heart TFBS and then everything forms from there. 2) just attracting new TFs, not particularly those of the "core" heart set, but some species-specific TF (at least species-specific in terms of expression domain).

Focusing on conservation is interesting, and chicken vs mouse hearts have surely a lot in common. But they also have a lot not in common. So new traits might arise from new enhancers. And this is what I was asking, are NC+ gaining TFBS for TFs that are different across species? which ones? are they interesting or just a not very informative mix of zinc finger motifs? Lack of conservation suggest that, but are the divergent TFs interesting in any way? In sum, are NC+ where we should focus our efforts when trying to understand lineage-specific adaptations? This said, DC/IC enhancers surely also contribute to species variation (tinkering with old tools) but perhaps de novo enhancers have a different story to tell. I just make this clarification in case the authors find the answers to this question relevant to be included in this manuscript, but I am fine if this is left for later studies.

Reviewer #3 (Remarks on code availability):

The github is well documented. My only suggestion here is that the tool is very focused to genomes that are available in UCSC, with named reference assemblies. However, in this day an age, single laboratory custom genomes are going to be more and more frequent. Also, UCSC is quite vertebrate biased, and given projects such as the Darwin Tree of Life, ERGA, etc, making this easily applicable to other lineages would be nice. Perhaps including the option (or some links to a guideline) to start with genome fasta files would be more broadly useable for the community.

Point by Point Response to Reviewers' Comments

NG-A65043, Phan, Zehnder et al.: "*Conservation of Regulatory Elements with Highly Diverged Sequences Across Large Evolutionary Distances*".

We would like to thank the Reviewers for the constructive and encouraging feedback and suggestions to improve the quality of our manuscript. We appreciate their assessment of the potential impact of our "*strong study*" and its presentation as "*well-written, clear in its analysis, and sound in its conclusions*".

We hope we will convince you that the significant changes, new data and new analyses added to the manuscript address all of the concerns raised. In particular, we have added a series of new analyses, additional experimental validations, and made substantial improvements on the IPP description and tool documentation to the benefit of the comparative genomics and regulatory genomics communities.

Based on the reviewer requests we have carried out the following key modifications and additions

- A more elaborate comparison of IPP with alignment based approaches (LiftOver) and improved description of the conceptual innovation of the IPP algorithm
- A complete IPP analysis of the published data from the "landmark paper" Villar et al 2015 that compared mouse to opossum liver enhancers.
- Addition of *in vivo* enhancer reporter validation of IPP-identified enhancer pairs (two heart enhancer pairs from less well characterized loci)
- A completely reworked documentation of IPP's github page, as well as now publicly available pairwise alignment files for our set of bridging species. This way, any already existing dataset of mouse, human or chicken CREs can easily be projected to any of the 16 species in our compendium with minimal computational effort.
- We reorganized the main figures. The former Fig. 2 is now split into two separate figures to give more room to the conceptual innovation of the IPP approach (new Fig. 2) and highlight the degree of functional conservation (new Fig. 3). Additional analyses are presented in revised main and 3 additional Supplemental Figures.

We find that the additional analyses, results and revised presentation have greatly strengthened the depth and scope of our study and we're looking forward to Reviewers' assessment of our revised version. Please find our Point-by-Point responses below:

Reviewer #1:

*Based on our new analyses and results, we restructured the main and Supplemental Figures. All references to Figures and text in the revised manuscript are highlighted in **bold**. When responding to comments addressing the original Figure numbering, these Figure references are not highlighted.*

Remarks to the Author:

“While the methods are robust and rigorously deployed, the study does not adequately explore the evolutionary relationships of IC enhancers or test to see if this model holds at later developmental stages (or other tissues), when functional conservation has been described to be minimal.

Response: We appreciate the Reviewer’s positive feedback on the quality of our data and the potential impact of our findings for the community. In response to the reviewer’s criticism, we provide new analyses of the sequence conservation of IPP projections, by which we demonstrate improved detection of evolutionary relationship between IC enhancers. Moreover, we feel that this new analysis also better demonstrates the conceptual innovation and effect of IPP through assessing positional conservation of regulatory elements. We also provide new analysis results of positional conservation of adult liver enhancers between mouse and opossum (based on enhancer mapping data from Villar et al 2015 PMID: 25635462). This new results demonstrate how the IPP tool can successfully and substantially increase the detection of putative orthologs by IPP in another tissue and indeed at a different ontogenic stage from the developing heart, which we have primarily focused on in our original MS. Please see further details in our responses below.

Overall, I think this is a strong study with a lot to offer to the community and that my criticisms should be addressable without significant new experiments/cost.

1.1 The authors very briefly describe what is required for DC status via UCSC LiftOver. While this is a fine choice, as liftOver is widely used and enables simple yes/no calling, this is a fairly simple way to look at sequence conservation. It would be of interest to test how relaxing alignment parameters (e.g using smaller intervals or changing from minmatch=.1) or using different alignment approaches changes how direct sequence conservation to be captured.

Response: We thank the reviewer and agree that a more extensive assessment of the sequence conservation status of DC and IC projections helps to outline the difference of IPP vs. common alignment-based approaches. The halliftover/HALPER analysis already started to explore this direction, but we added further analyses to dive deeper.

- We expanded the liftover analysis by parameterizing the minMatch threshold and report the results in **new Fig. S3a**. Relaxing the minMatch threshold does not lead to a significant increase in sequence conserved regions. Therefore, we do not expect that the majority of “paired IC/NC sequences map using other sequence homology methods”.
- The results show that although DC regions identify a similar number of orthologs, they are not interchangeable datasets. In fact, ca. ~50% of the LiftOver orthologs are DC and 30-40% are IC, with a minority representing LiftOver unique hits (**new Fig. S3a, b**).
- Mouse enhancers projected to the chicken genome by IPP show increase evolutionary conservation at these sites as measured by PhastCons77 scores (**new Fig. 2d**)

These results highlight the conceptual difference between the positional conservation of an IPP projection and the sequence conservation of current approaches. We describe changes in the revised paragraph “IPP improves detection of orthologs between distantly related species” (**line 144-157**). Since in our study different types of evolutionary conservation are compared (sequence conservation,

positional conservation, functional conservation), we clarified these differences in the text by specifying which flavour of conservation we refer to wherever appropriate.

1.2 Related to the above point, it would be of interest to do a deeper dive on the homology that is present for the IC class. It seems that the synteny model would be that sequences that hit IC thresholds should be largely descended from common ancestral sequence?

We fully agree with this interpretation. Through positional conservation, IPP can detect orthologous sequences too divergent to be alignable. We included the phastCons77-way score for the DC/IC/NC projected CREs in the chicken genome in **new Fig. 2d**, which shows that DC and IC projections are enriched for evolutionary conserved regions, albeit the signal for IC is expectedly lower than for DC.

1.3 Or could these be insertions of sequences that are not presumed to be evolutionarily shared?

Because IPP is based on synteny, IC regions are unlikely to be stemming from insertions (such as retroelement insertions/sequence-similar regions that may be present at several genomic locations). This scenario would expect/imply lower-stringency sequence homology. However, since it would not reside in the same chained alignment that is the basis of IPP's synteny approach it would not be picked up by IPP. Therefore, we do not conclude that IC regions could stem from "insertions of sequences that are not presumed to be evolutionarily shared".

1.4 As an example, do paired IC/NC sequences map using other sequence homology methods that might permit reduced alignment (e.g. BLAST), or are they so divergent as to be unalignable or stemming from retroelement insertions?

Our expanded LiftOver analysis (see 1.1) does not suggest so. We also performed BLAST for all IC pairs tested *in vivo* - none were alignable.

Without deeper investigation, the methods advancement is somewhat murky as the IPP methods are already published and the biological relevance is muted as the new insights are limited with regard to the origins and make-up of IC enhancers.

We think that these additional analyses help to exemplify the innovation that the positional conservation detected by IPP represents. However, to leverage the potential of IPP required the combination of matched experimental data with the algorithmic innovation of IPP. This allowed us to quantify sequence-conserved vs. positionally conserved enhancers, which we feel represents a significant advancement of our study.

We also completely reworked and **updated the github repository** to make IPP an easily accessible computational tool. The new analyses on the evolutionary conservation of IPP projections are presented in **new Fig. 2** and **new Fig. S3**, and in the revised paragraph "IPP improves detection of orthologs between distantly related species" (**line 127-176**).

2. More analysis should be done on comparing the three enhancer classes (DC/IC/NC) – the current paragraph is not very informative. The present manuscript states similar chromatin state (H3K4me1 at enhancers and H3K4me3 at promoters) and overall similar chromosomal contacts. But the authors should have the data to dive deeper.

Response: We apologize for this omission. After implementing IPP on our heart CREs, we initially checked whether in the mouse (i.e. source) genome DC enhancers differ from IC/NC enhancers based on commonly used enhancer marks and other functional data, but did not find any unexpected differences. IC/NC enhancers/promoters appear as just as fine promoters or enhancers as DC (based on chromatin marks and TFBS). In addition, we also compare the distribution of enhancers relative to genic features.

What is the distribution of DC/IC/NC enhancers with regards to an average genomic interval?

DC/IC/NC enhancers are similarly distributed relative to TSS's (**new Fig. S4c**)

Is there a difference in presence of H3K27ac or in strength of histone ChIP-seq signal?

DC/IC/NC CREs do not differ from each other in chromatin signatures (**new Fig. S4a**)

DC/IC/NC promoters and enhancers are equally likely to be bound by multiple heart TFs (ReMap 2022 data) (**new Fig. S4b**)

Is there a difference in HiC/contacts or TAD positioning for DC/IC/NC loci? For example, are DC/IC more or less interchangeable in the average number per gene/locus? Are NC enhancers more likely to be found near DC/IC enhancers?

In addition to the observations above, we detected that DC & IC enhancers are more likely intragenic than NC, but they have different proximities to exons. This distribution is expected due to the way IPP works (**new Fig. S4d**). For evolutionary comparison of Hi-C interaction patterns please see Reviewer 2 Minor Comment 4.

We describe all results mentioned above in a new paragraph prior to exploring the chromatin signature at the projected regions in the chicken genome. (**I 202-213**)

3. IT would be of interest to know how the DC/IC/NC proportions for annotated enhancers varies across tissues/developmental stages. As the authors point out, once the projections are done, it is easy to extend to other regulatory loci sets (as done for liver TF ChIP and limb enhancers in the manuscript).

Response: We agree with the Reviewer that this is an interesting and impactful comparison to provide. We decided to reanalyze the data from Villar et al 2015 PMID: 25635462, because it was useful for a number of reasons as well as addressing a similar point raised by Reviewer 3 (Major 2):

- It represents a different developmental stage in an unrelated organ (adult liver)
- It was created to characterize orthologous enhancers based on chromatin modifications
- It reports conservation to the relative distantly related opossum, representing an evolutionary distance where IPP is particularly useful
- The ChIP-seq data in opossum, although different to ours, allowed us to run a similar downstream functional comparison

We reprocessed the ChIP-seq data from Villar et al and identified putative CREs (the published peak set did not match our peak-calling criteria and genome annotation), and projected enhancers and promoters to all species in our comparison. We find a similar increase of putative orthologs for IPP (e.g. 5-fold increase of enhancers to chicken, 4,921 new putative orthologs). With now two sets of embryonic heart enhancers, embryonic limb enhancers, adult liver enhancers and CEBP/A binding sites, we have a consistent increase in ortholog identification with IPP, that shows ~3- and ~6-fold increase for promoters and enhancers respectively (for mouse-chicken).

We also used the availability of the mouse and opossum ChIP-seq data to run a similar classification of functional conservation for mouse-opossum liver enhancers, which allows for further overarching assessments of enhancer evolution. Functional conservation (DC+/- and IC+/-) was slightly different in numbers, but overall consistent with our heart and other published data (Viestra et al. 2014, see also Reviewer 3, Major 3).

IT would be of interest to compare more broadly, e.g. to some of the mouse vs. human tissue samples in ENCODE. This would increase the impact and enable some overarching comparisons about enhancer conservation.

We see a consistent increase in the number of putative orthologs that IPP achieves, with the absolute numbers varying between embryonic limb (e.g. 13,3% DC vs. 49,8% DC+IC enhancers mouse-

chicken), heart (8,8% DC vs. 43,3% DC+IC), and adult liver (6,2% DC vs. 38,9% DC+IC). However, such differences as well as those in functional conservation mentioned above can stem from a number of sources: data availability (two chromatin marks for liver vs. three + ATAC-seq in heart); difference in biological sample (adult liver vs. embryonic heart); different evolutionary distance.

Therefore, we are cautious with drawing too grand conclusions, but state that the degree of functional conservation is lower than often assumed (please see also Reviewer 2 Major 2 & Reviewer 3 Major 3)

We incorporate the new liver enhancer data in **new Fig. 3** and **Fig. S3** and at two positions in the Results section:

- Description of ortholog detection in published data (including Villar et al (**I 179-188**))
- Identification of functionally conserved CREs in Villar et al (**I 229-239**)

We also added a specific Discussion of the low degree functional conservation (**I 412-421**)

We are very happy for receiving this impulse to expand our analysis and we feel that these analyses and results broaden and deepen the scope of the study. Thank you for the suggestion!

4. Minor point – it is unclear how the two timepoints are used. In some places, both timepoints are stated (E10.5/E11.5) compared to others where only one timepoint. The supp fig shows some presumed stage-specific differences. How do DC/IC/NC stats compare for the timepoint specific vs. shared loci? The manuscript should clarify when and how the data is used for the multiple timepoints.

Response: We obtained data from the two timepoints to ensure that we cover small differences in the exact developmental timing between the two species. All initial IPP analyses were performed for E10.5/HH22 and E11.5/HH24 independently, but there were no differences in conservation/enrichment patterns, which is why we decided to run the major analyses of the paper on the union set of enhancers and promoters to not complicate the manuscript.

All signal enrichment plots and the comparison of ATAC-seq peak conservation is stage-specific data from E10.5/HH22 (E11.5/HH24 analysis look pretty much identical, which is why we decided not to show this set of analysis). We see that the Venn diagram in Fig S1 has not helped with providing the intended clarity.

We adjusted the main text (**I 93-95**) and **revised Fig. S1** to make this distinction clearer.

Reviewer #2:

*Based on our new analyses and results, we restructured the main and Supplemental Figures. All references to Figures and text in the revised manuscript are highlighted in **bold**. When responding to comments addressing the original Figure numbering, these Figure references are not highlighted.*

We thank the reviewer for sharing the excitement for our study and understand from the Major Concerns the need to improve the description and conceptual innovation of IPP, as well as the validation for IPP-unique predictions. To do so, we performed several additional experiments and analyses as outlined below.

The authors compare their method to other alignment-based methods using hierarchical trees, and conclude that their method performs as similarly or better at relatively short evolutionary distances, but has a clear advantage over large distances (Lines 142), as it identifies more putative orthologous enhancers. The key question then is are the regions that they discover in distally related species really new enhancers?

Response: We hope to convince the Reviewer about the main achievement of our study not necessarily being the detection of new enhancers, which we believe are best explored by biochemical and genetic tools, but to offer an efficient and reliable approach to investigate the evolutionary trajectory of enhancers at varying evolutionary distances and at varying rate of change in genomes. We addressed the fundamental problem of detecting homologous and functionally equivalent enhancers and how to assess their relationship when conventional sequence alignment approaches fail.

We thus do not expect that the majority of orthologs that IPP identifies are “new” enhancers. Instead, the conceptual advance is that our approach allows to determine ortholog enhancer pairs without relying on homologous sequences. Enhancer conservation and divergence has been proposed to underlie speciation, yet our ability as scientific community to generate hard data on the scale, extent and impact of enhancer conservation remains sparse due to the rapid divergence of regulatory sequences.

We think that our tool will help the evolutionary genomics community in demonstrating orthology of sequence-diverged enhancers and in decoding the semantic rules of enhancer sets. We offer an important step in this endeavor by presenting comprehensive strategy for detection of positionally conserved CREs that descended from a common ancestral sequence and diverged so much that they lost alignability while retaining function.

1) In vivo testing in transgenic embryos – which is the definitive proof.

It's great that the authors tested some chick IC enhancers in mouse – But unfortunately, the four IC regions tested were biased – selected around the four major heart lineage genes, Hand2, Tbx20, Nkx2.5, Gata4 – these are the main conserved lineage factors in all species for the heart. If one just took chick ATAC peaks with H3K27ac signal in the vicinity of these genes, there is a very high probability that they would also give heart activity in mouse. The authors didn't show that the IPP gave any advantage here. Also, how exactly were the tested regions selected, they just state “we selected two pairs of DC and 4 pairs of IC enhancers.” – no information is given in the text or methods. What distance to the genes was used?

Response: As noted above our primary aim was to detect the evolutionary relationship of enhancers. As *in vivo* enhancer testing in mouse cannot be done at scale (except for a setup like the Pennachio/Visel labs), our selection of tested enhancers needed to be limited. In this context the chosen loci represented well-founded models as they offer reliable context in which *in vivo* testing could be interpreted.

However, the argument regarding enhancer peaks in the vicinity of main lineage factors is well taken. While our initial selection of enhancer pairs had some of the points raised in mind, we realize from the Reviewers comment that this motivation was not explained, and any description of the enhancers selected was too brief.

- We chose DC and IC enhancer pairs at different distances to main lineage factors
- In the case of Tbx20 we compared one DC and IC pair

- We included an intronic IC enhancer pair of a gene that is not extensively described in the literature (*Pakap*).

We therefore rewrote the entire section which now describes our motivation for the selection of enhancer pairs tested *in vivo* (line 302 - 330). We also created a **new Supplemental Fig. S6** that shows genome browser screenshots of the selected loci and enhancer position to the presumed target genes.

Did you use Phastcons or other signatures of conservation or selection? Is there a confidence score from the IPP model? If yes, was this used?

IPP produces an IPP-score that reflects the confidence of IPP's projection. To be IC, the IPP score must be above 0.875, which corresponds to a summed distance of <2.5kb to all corresponding anchor points (at every intermediate projection). We revised the main text (I 106f) and Fig. 2 (now split to Fig. 2 and Fig. 3) to better describe what IPP does and what DC/IC/NC signifies. We also improved the detailed description of the IPP method and score in a revised Supplemental Text (I 582f).

To really show the advantage of the method – take some regions from poorly studied genes, or regions very distal that are not obvious where you could only find them with the IPP method and not by sequence alignment and/or chromatin signatures. This would greatly strengthen the manuscript.

In addition to the *Pakap*-IC enhancer pair, we expanded our set of heart enhancers tested *in vivo* by testing 2 enhancer pairs from within introns of non-heart specific genes: *Mig1a* and *Auts2*. Both murine enhancers are smaller fragments of previously characterized heart enhancers (*mm72* and *mm131*) Vista Enhancer Database) in the mouse and offered an ideal opportunity to seek orthologous elements in a distant species. As the reviewer might expect, no ortholog chicken enhancers have been detected before. Importantly, our results demonstrate that the IPP-predicted ortholog chick enhancers drive heart-specific expression (Fig. 5) and suggest them as the sequence-diverged homologs to mouse counterparts detected by IPP.

While the number of elements we tested are minimal and no way statistically significant, nevertheless we now summarize conserved expression driven of the enhancer pairs tested in a **table in Fig. S7d**. We would argue that comprehensive and statistically relevant large-scale testing of candidate orthologs is beyond a single lab's capacity.

However, we think that our results are already very promising for IPP as we validate 4/6 IC heart enhancer pairs as functional orthologs despite absence of any sequence homology, which should encourage the comparative genomics community to test positional conservation in their studies, to whom we offer our functional conservation prediction tool.

In the manuscript, we revised the results section (I 302 - 330), revised main figure (**new Fig. 5**) and new supplemental **new Fig. S6, 7** to describe the enhancer pairs tested and the outcome. We think these examples for positionally conserved enhancers will enable the reader and the community to fairly assess the advantages of IPP.

2) A more global assessment of IC vs DC enhancers with chromatin signatures:

Line163 – As only the coordinates of the regions experimental identified (using H3K27ac, ATAC data) in mouse were used by IPP to identify the regions in chicken, the authors could use the chicken chromatin data as validations – if the IPP IC regions are correct, they should have genomic signatures of enhancers and promoters, which all makes logical sense.

The results were good on the one hand, as they were comparable between DC and IC enhancers For DC: 66% promoters and 29% enhancers had an ATAC peak in chick. For IC: 56% promoters, and 26% enhancers (line 170) This is in line with each other – but even the percentages for the direct conserved regions are surprisingly low for enhancers – 30% are bound (ATAC peak). How does this compare to other methods, e.g. the hierarchical method mentioned?

Response: Yes, we were also initially surprised by the relatively low number of functional conservation (especially of DC CREs) as also pointed out in by Reviewer 3 Major point 3.

However, this number is fully in the range of what has been reported previously for sequence-conserved peaks of the ENCODE datasets comparing corresponding mouse and human samples (Viestra et al 2014 PMID: 25411453). Another study compared epigenetic conservation amongst primate LCL CREs, and only 41% of ChIP & ATAC defined strong (sequence-conserved) enhancers are functionally conserved in LCLs among all 5 great apes (Garcia-Perez et al 2021 PMID: 34035253). Given that both publications compare a much closer evolutionary distance than our mouse-chicken comparison, our data does not appear unreasonable.

In the revised manuscript we highlight this by including an analogous LiftOver +/- analysis in the **new Fig. 3b/c**, which shows that only a fraction of sequence-alignable enhancers share functional conservation.

Furthermore, our reanalysis of the Villar et al liver enhancer data shows the DC+/IC+ fraction of all DC/IC enhancers are only 32% (DC) or 18% (IC) (lower than observed for our mouse heart enhancers in chicken), the promoter conservation in these data is higher. However, given the difference in input data, compared species, bioinformatic enhancer calling etc., we do not want to speculate about the absolute percentages that can be expected for defining functional conservation.

From this and Reviewer 3's comments we realized that stating this relatively low number of functional conservation is an important result to highlight from our study. Although it has been described in the past (Viestra et al 2014 PMID: 25411453) for mouse-human DHS, it is rarely stated explicitly. Such different usage of conserved enhancers species might be an underexplored evolutionary mechanism. Therefore, we also added a new section in the Discussion (**I 412-421**):

“Sequence conservation of CREs, especially that of enhancers, displays a great level of heterogeneity ranging from ultra-conserved elements (Snetkova et al. 2021; Dickel et al. 2018; Snetkova et al. 2022) to the sequence-divergent IC elements we describe here. **One notable observation is that especially for enhancers, functional conservation in chromatin state (i.e. DC+/IC+ elements) accounts a surprisingly small fraction, despite our efforts to obtain high quality CREs from developmentally equivalent tissues. This is not only true for IPP projections, but also for LiftOver orthologs. Our data, however are consistent with adult liver enhancers (Villar et al. 2015) reanalyzed here and similar to the functional conservation of sequence-conserved DHS sites between human and mouse, which were reported for equivalent cell types obtained in the ENCODE consortium (Vierstra et al. 2014). This suggests that alternative activity of ortholog enhancers might be more widespread than currently appreciated and functional conservation, in terms of chromatin signatures, encoded TFBS, and predicted tissue-specificity is relatively uncoupled from sequence conservation.**”

3) A more global assessment of IC vs DC enhancers by motif analysis.

Using gkm-SVM – IC enhancers were equally likely to be classified as heart enhancers as DC enhancers (AUC ~0.75 for each), providing good evidence that there is a global signature there. Although I don't agree with this statement “known motifs of master regulators of heart development (e.g. GATA, TEAD and HAND) were most predictive of tissue specificity (Fig. 3c)” – line 211, At least the data is not shown in Fig 3c – only the PWMs for the motifs. Show a ranked ordered heatmap of the p-values of motifs, so we can see which are the most predictive.

Response: We apologize for this omission. The original Fig. 3c and Fig S4 e+f (**now Fig. 4c and Fig. S5 e+f**) showed ranked lists of informative sequence patterns (seqlets) and matching PWM hits. We realized that we did not include any quantification or description of how the ranking was determined. The seqlets are results from de novo motif discovery using TFMoDisco, showing clusters of patterns with high contribution scores from all sequences that the SVM predicted as "heart-specific". Analogous to (de Almeida et al 2022, DeepSTARR PMID: 35551305), we ranked the motif patterns by how many seqlets are found for a particular pattern and now present this information as “# of seqlets” in **Fig. S5e+f** analogous to the Supplementary Figure from de Almeida et al 2022)

Only the functionally conserved CREs (i.e. with chromatin marks) have the same motifs for DC and IC regions. However, IC+ enhancers shared as many motifs as with DC- regions - Line 225. How do you explain that? It does not fit with a 'shared syntax to DC+ enhancers.

We assume that sequence conservation in DC does not depend on the heart TFs alone, but possibly is due to pleiotropic enhancer activity and motif similarity in cells where that CRE is active in another tissue (E.G. GATA3 vs GATA4 motifs), which is supported by our analysis in **Fig. S5d** (former Fig. S3d). This is described in **I 256-261**.

There is little/no evidence for a conserved motif syntax etc for the IC enhancers, as it states in the text.

The reviewer is correct, we referred to the number of shared TFBSs (which is higher in DC+ vs DC- and IC+ vs. IC-). The fact that these are more shuffled as shown later in the manuscript speaks against a conserved syntax. In that context, the phrasing we use “conserved sequence syntax that is independent of direct sequence conservation“ is paradoxical.

We revised the sentence (**I 298-300**) to:

“These results confirm that IPP identifies orthologous pairs of CREs with shared TFBS, representing conserved regulatory information that is independent of direct sequence alignability.”

While I appreciate that the two global assessments discussed above (based on chromatin and motifs) [...]. However, neither of these analyses were done with other methods (e.g. halliftover/HALPER). Which enhancers did this method identify that IPP didn't?

In response to Reviewer 1 Major 1 we added a more extensive exploration on the conservation of IPP projections that helps to address the issues raised here.

DC/IC/NC definitions are not directly dependent on sequence homology (only indirectly via nearby anchor points). That means that DC regions, for example, overlap but are not interchangeable with LiftOver orthologs. This is now shown in **Fig. S3a** and corresponding text (**I 144-158**). While IPP finds many more regions in addition to LiftOver (also if minmatch thresholds are reduced), LiftOver barely detects regions that are not included in IPP projections (**Fig. S3a,b**).

Do those enhancers also have chromatin and motif signatures suggestive of heart enhancers?

As discussed in the above (Major Comment 2), we performed an equivalent LiftOver +/- analysis shown in the **new Fig. 3c+d** that also sequence alignable CREs are only partially functionally conserved based on chromatin signatures. This is in the same range as for IPP-projected CREs, even though these are 5-fold higher in number.

In other words how well is the IPP method performing to identify real functional enhancers compared to other methods?

Our results show that IPP identifies the vast majority of CREs that can be identified with alignment based methods but adds 3-5 fold more putative orthologs (mouse-chicken/promoter-enhancer). The functional analysis based on chromatin marks demonstrates that that there is additional variability of the chromatin signature that is relatively decoupled from the sequence conservation. Therefore, we believe that IPP provides a very intriguing tool to increase the detection of putative orthologs in evolutionary comparisons. However, validation of functional conservation requires additional experimental data, which we (for example) provide with the heart chromatin profiles.

We hope that the additional analyses, new results and textual changes throughout the manuscript help to clarify these insights from our study.

4) The authors provide the source code in Github, which is great. But as this manuscript is primarily a new computational method, to be generally useful and a significant advance, the authors should provide the IPP method as a user-friendly package (in BioConductor or elsewhere).

Response: We fully agree with and have used this opportunity to implemented **substantial updates to our documentation on the github repository** (<https://github.com/tobiaszehnder/IPP>), which should now provide and easily accessible tool to use for the evolutionary/comparative genomic community.

Specifically:

- We have added a step-by-step instructions for executing IPP, including example usage and what the required input file are and how to obtain them.
- We provided the large collection of alignment files necessary to run IPP by depositing them on a publicly available web server (<http://owwww.molgen.mpg.de/~IPP/>). These include

- alignment files for three major comparisons: human-all, mouse-all, and chicken-all using our optimized set of bridging species for mammalian-reptile comparisons.
- We provided a pipeline to generate these input files for researchers looking to use their own species collections.
 - We offered detailed explanations of the output files generated by IPP: what they look like and what information they provide.

The pairwise alignments for our set of bridging species optimized for mammalian-reptile comparisons already provides the necessary alignment files required for projecting any enhancer set from three major model species (human, mouse, chicken). This way, such comparisons can be readily performed with minor pre-processing steps.

A researcher (Jacob Hepkema from Leopold Parts lab at the Sanger Institute) who contacted us following the preprint's release, was kind enough to test and give independent feedback on the revised github and resources.

Minor comments

1) How was the equivalent developmental stages defined between chicken and mouse?
This is central

Response: We chose the timepoints with regards to equivalent developmental stages, not only for overall embryological development, but for heart formation in particular. These midgestation stages are selected towards the end of the process of heart looping with all future chambers visible, but not yet fully formed. The stages are very similar between mouse and chicken and were also chosen to obtain sufficient material for functional genomic studies.

This process has been anatomically described in detail in mouse and chicken and we realized that we did not reference key publications. This has been addressed in the revised manuscript (**I 75**).

- | | | |
|----------------|---|--|
| Martinsen | - | Reference Guide to the Stages of Chick Heart Embryology; Development 2005 |
| Moorman et al. | - | Development of the heart: (1) Formation of the cardiac chambers and arterial trunks; Heart 2003 |

2) What was the definition used for sequence conservation? For example, at line 94/95, and line 130 – (Fig 2C) 22% promoters, 10% enhancers are conserved – the definition used to define something as conserved should be stated in the main text – this is crucial

Response: Thank you for pointing out this oversight. Given the content of the manuscript we were careful to call regions sequence-conserved (i.e. (LiftOver (--minMatch = 0.1)) when we did just that. The distinction between sequence conservation and IPP-projections should now be clearer due to the added analyses in response to Reviewer 1 (Major 1). We revised the specific sentence to avoid confusion (**I 137**).

3) What is the global conservation of TADs? Fig S1

Response: The “global conservation of TADs” can be measured in many different ways and comes with a number of technical and biological challenges. One way is to follow the analysis of the 2017 Harmston et al. paper, which is often cited as a validation of TAD conservation. This analysis shows that ancient boundaries of human:chicken GRBs overwhelmingly correspond to TAD boundaries. These prototypical TADs – frequently encompassing developmental gene deserts – are generally well conserved.

In addition to the synteny breakpoint analysis in **Fig. S1c** we now include a new analyses of our Hi-C data which we combined with GRBs defined in the mm10 and galGal6 genomes. Analogous to Harmston et al. 2017 we show the conservation of GRBs and TAD boundaries in the **new Fig. S1d**.

- 4) Line 185 – give the reference for the gapped-kmer model
Line 186 “we first trained a gapped k-mer Support Vector Machine (gkm-SVM) model on mouse data to identify heart-specific enhancers. “
The model is not identifying heart enhancers from the rest of the genome de novo - you are providing it with the heart enhancers. It is rather learning the k-mers within heart enhancers – this sentence should be changed or merged with the next one

Response: We included the references (which we had cited in the Methods) and adjusted the text as suggested.

- 5) Fig 3c, it is surprising that Nkx2.5 motif is not present.

Response: We agree, it might be intuitive that the Nkx2-5 should be detected by the ML model. However, within the cardiac GRN, Nkx2-5 acts upstream of most other factors with the exception of Gata4, which is a pioneering TF. Accordingly, Nkx2-5 plays an important role during early morphogenesis, most particularly during the linear heart tube stage (~E8.0 in mouse), where its loss resulted in embryonic lethality with a heart tube failing to undergo looping (see Lyons et al, 1995 PMID: 7628699). Since our model focused on later stages of heart development (i.e. E10.5, E11.5, E12.5, E14.5 and P0), it is not entirely surprising that we did not detect the Nkx2-5 motif from heart-specific enhancers as classified by the SVM model.

Alternatively, why is the CTCF motif enriched?– it should have been cancelled out in the background.

It was indeed unexpected that the CTCF motif was enriched given such a background. Nevertheless, the occurrence of CTCF motifs is only one among many detected signatures driving the positive prediction of an active heart-specific CRE. It is feasible that co-occurrence with other motifs is what drive tissue-specificity, but further investigation into such features relating to motif syntax and enhancer grammar would require more sophisticated ML-models and is beyond the scope of this manuscript.

How was the background matched for the gkm-SVM predictions in heart enhancers? Was another set of enhancers used? Or all other distal ATAC peaks?

To train our SVM heart model, the negative sets were non-overlapping peaks (at least 2kb away from positive peaks) obtained from candidate regulatory elements as annotated by ENCODE (cCREs) and ATAC-seq peaks from non-heart embryonic tissues (limbs, mid-/fore-/hind-brain, and liver). Additionally, these regions are GC- and repeats-matched to have the final negative/background set consisting of putative regulatory regions without known activity in the heart.

Reviewer #3:

*Based on our new analyses and results, we restructured the main and Supplemental Figures. All references to Figures and text in the revised manuscript are highlighted in **bold**. When responding to comments addressing the original Figure numbering, these Figure references are not highlighted.*

1- The pipeline for IPP is deposited in GitHub, and that's great, but I have been looking at the README file and it does not look very well explained. In contrast, the github repository of HALPER is very detailed. Since the IPP approach would be a main sell of this work, as many groups might want to try similar things with their own set of genomes, I think the pipeline should be much more detailed, with a step-by-step explanation on how to obtain the input files, how to process them, how to run this approach, and what the output files look like. "python project.py" is the only command that is specified in the repository, which kind of speaks for itself. The authors cite the DANIO-CODE paper (Baranasic et al 2022) as a reference, but there I couldn't find a tool description more detailed than in the github associated with this repository. If the main advantage of this method is that it requires less input sequences than HALPER and is computationally less intensive, a widely usable code would increase the method's adoption by the community and the impact of this work.

Response: Thank you for your feedback and for giving us the opportunity to improve the usability and accessibility of our resources.

We fully agree with your suggestion and have implemented **substantial updates to our documentation on the github repository** (<https://github.com/tobiaszehnder/IPP>), which should now provide an easily accessible tool to use for the community evolutionary genetics.

Specifically:

- We have added a step-by-step instructions for executing IPP, including example usage and what the required input files are and how to obtain them.
- We provided the alignment files necessary to run IPP using our set of bridging species by depositing them on a publicly available web server (<http://owwww.molgen.mpg.de/~IPP/>).
- We provided a pipeline to generate such input files for researchers looking to use their own species collections.
- We offered detailed explanations of the output files generated by IPP: what they look like and what information they provide.

The pairwise alignments for our set of bridging species optimized for mammalian-reptile comparisons already provides the necessary alignment files required for projecting any enhancer set from three major model species (human, mouse, chicken). This way, such comparisons can be readily performed with minor pre-processing steps.

A researcher (Jacob Hepkema from Leopold Parts lab at the Sanger Institute) who contacted us following the preprint's release, was kind enough to test and give independent feedback on the revised github and resources.

2- Using the chicken versus mice comparison as the centre for the analysis is well justified, as heart enhancers have been particularly difficult to find using sequence conservation alone. Still, I think using an orthogonal dataset to some extent would also increase the potential impact of this work. For example, reanalysing Villar et al 2015 data, and highlighting the new numbers of conservation using the IPP pipeline would be impactful, as it might somehow overturn the quantitative expectations of the field (most enhancers are not conserved at the level of sequence, yet most are positionally).

Response: We thank the reviewer for this encouraging comment! Our initial submission started in this direction by including the Blow et al and Schmidt et al data (since they either focused on heart or had a mouse-chicken comparison) in addition to a limb CRE set we had created.

We took up the reviewer's suggestion and now expand on this through an extensive reanalysis of the Villar et al. liver ChIP-seq data set and provide orthogonal analyses as much as that data allows.

- We reprocessed the ChIP-seq reads from Villar et al and identified CREs (the published peak set did not match our criteria and genome annotation). We then projected these mouse CREs to all

species in our comparison. We find a similar increase of putative orthologs using IPP (e.g. 6-fold increase of enhancers to chicken), this data is shown in **new Fig. S3f**.

- Because the Villar dataset also allowed to do a similar assessment of functional conservation based on chromatin marks between mouse and opossum, we perform a DC+/DC-; IC+/-; LiftOver+/- comparison. The results are consistent with our heart-data, although slightly different in numbers. We think this is most likely due to a combination of data available (only two chromatin marks (H3K4me3 & H3K27ac)); lower evolutionary distance; difference in the biological sample (adult liver vs. embryonic heart). The results are shown in the **new Fig. 3c** (please see also response to Major Comment 3).

These new analyses (textual changes in particular **I 178-194** and **I 229-239**) highlight that positional conservation in fact is general and far more widespread than currently appreciated based on sequence conservation. We agree that this confirms the field's expectations. However, to our knowledge supporting experimental data has been lacking, not least due to absence of an approach like IPP.

The initial description of the bridging species approach (Taher et al 2011 PMID: 21628450), or another study aiming to identify divergent enhancers in syntenic introns (Yang et al 2015, PMID: 26519295), solely rely on sequence comparison and are not combined with a high-quality chromatin feature based enhancer data set as done in our study, and which has become the norm to identify regulatory regions.

Therefore, we believe the quantitative appraisal of positional/indirect enhancer conservation in combination with the now easily accessible computational tool IPP can have a significant impact. Our approach enables the field to systematically reevaluate the conservation and evolutionary trajectory of regulatory elements without relying on sequence-alignable regions as has been done in some prominent examples such as Villar et al 2015 PMID: 25635462, Viestra et al 2014 PMID: 25411453, Dickel et al 2016 PMID: 27703156, Berthelot et al. 2018 PMID: 29180706). Notably, those studies mentioned above address inter-mammalian enhancer conservation, possibly because sequence conservation rapidly drops at larger evolutionary distances (see LiftOver matches in **Fig. 2**)

By presenting strong experimental support for positional conservation of regulatory elements, we hope that our results will be impactful. We would like to thank the reviewer for the impulse to expand and deepen our analysis, which we feel have substantially strengthened the manuscript.

3- Perhaps the most shocking result that is not really discussed are the DC- enhancers. These are sequence conserved, but not equivalently active in the chicken-mouse comparison. When analysing the TFBS composition, the IC+ (positionally conserved active in heart in both species) show a similar level of conservation of TFBS regulatory logic to DC-. Perhaps these DC- enhancers are pleiotropic and that's why they are conserved at the level of sequence but have suffered some change that makes them not active anymore in the heart of either species. But since they have an equivalent composition of TFBS, then what changed? Perhaps looking at silencing histone marks (probably already available for mice at least) could explain this divergence in function despite sequence conservation. Also, it would be important to see what patterns of expression some DC- enhancers from chicken drive in mice. Despite the higher sequence conservation, are they no longer capable of driving heart expression patterns? Or is a change in trans unique to chickens that silences the enhancers despite harbouring a regulatory lexicon capable of heart expression?

Response: We fully agree with the reviewer and our initial reaction to the data was similar. However, when comparing with the literature, our numbers fall well within the range reported previously for sequence-conserved peaks in ENCODE datasets comparing corresponding mouse and human samples (Viestra et al 2014 PMID: 25411453). Similarly, Garcia-Perez et al. (2021, PMID: 34035253) analyzed functional conservation among primate LCL CREs and found that only 41% of CHIP- and ATAC-defined strong enhancers (sequence-conserved) were functionally conserved across all five great apes. Given that both publications compare a much closer evolutionary distance than our mouse-chicken comparison, our findings appear consistent with the literature.

While low functional conservation is presented as major observation in the Viestra et al 2014 PMID: 25411453 publication, this is not explicitly stated in numbers in the more sophisticated bioinformatic analyses in Villar et al 2015 PMID: 25635462.

To substantiate this point, we included two additional analyses shown in the **new Fig. 3b+c** and reported in **I 219-239**. First, to show that the low functional conservation is not due to IPP as a novel approach to identify orthologs, we performed an equivalent LiftOver +/- analysis for the murine heart enhancers. This clearly shows, that also sequence alignable CREs are only partially functionally conserved based on chromatin signatures. Thus, IPP-projected CREs, even though 5-fold higher in number, identify functionally conserved enhancers with comparable robustness. As mentioned above (Major Comment 2), we now performed an equivalent functional conservation analysis of mouse liver enhancers in opossum (**new Fig. 3c**) and find similar (LiftOver & DC) or slightly lower (IC) functional conservation (based on chromatin marks) than in our mouse-chicken heart enhancers.

We believe that differences in percentages are at least partially due to technical reasons (available chromatin marks, evolutionary distance, precision of projections) and therefore do not want to make to far-reaching conclusions regarding differences between liver and heart enhancers.

A full analysis exploring the reasons for this turnover of chromatin activity would substantially divert and overburden the manuscript and clearly warrant its own paper. In this study we emphasize the quantity of positional conserved enhancers and challenge the assumption that most enhancers are not conserved.

Regardless, we fully agree that this surprisingly low functional conservation of enhancers in itself is an important observation to highlight explicitly. Therefore, in addition to the reanalysis of the Villar-data we also added an according paragraph to the Discussion (**I 412-421**).

4 - Then, the authors should also do a motif analysis on the NC+ set of enhancers. Are these likely “novel” enhancers showing a different regulatory logic than the DC/IC? The TFBS conservation seems lower, but are the same motifs that dictate the general signal for the tissue also present there as the most enriched?

Response: The former Fig. 3e+f (**new Fig. 4e+f**) already includes a motif analysis and ATAC-seq footprinting for NC+ promoters and enhancers showing that as “uniform class” of projections they underperform compared to IC and DC elements.

Or the non-conserved enhancers are really the place to look for regulatory innovation?

Within NC projections, we expect a range of false negative “true orthologs”. Because IPP relies on distance to syntenic anchor points, its projection accuracy diminishes with larger distance to an *anchor point*. The definition of DC/IC/NC projections represents a trade-off between specificity and sensitivity and are determined by IPP score cutoffs (discussed below). Therefore, they should be thought of as high, medium, and low confidence projections. We adjusted the main text (**I 120-121**) and created the **new Fig. 2** to make this conceptual point clearer.

IPP computes a score for every projection, which is a representation of the distance to the nearest anchor points, i.e. the higher the score, the shorter the distance and thus the more accurate the projection. In fact, parameter tuning is an interesting feature of IPP that can be exploited according to the needs of the underlying data. The Supplemental Text is devoted to explaining the ways IPP can be used and therefore separate from Materials and Methods discusses this. But briefly:

DC regions: projections ≤ 300 bp from a direct alignment between the source and target genomes

IC regions: projections > 300 bp from a direct alignment or require a bridged alignment where the sum of distances from the query element to an anchor point at all intermediate projections is ≤ 2.5 kb

NC regions: projections with summed up distances > 2.5 kb from an anchor point through all intermediate projections

This means that the NC cutoff is somewhat arbitrary and we decided to remain more conservative in this study. Therefore we expect that NC+ projections in many cases represent projections close to the IC+ cutoff.

Then, I have some minor comments:

Line 218. “If IPP projections represent conserved pairs of CREs, these regions should share the same TFBS.” Well, not really, conserved CREs might also have some differences on TFBS, not all should be conserved (some for sure). Perhaps rewriting as it “should share more TFBS in common than non-conserved enhancers”.

Response: Yes. Adjusted to:

“If IPP projections represent conserved pairs of CREs, these regions should **have more TFBS in common than non-conserved enhancers.**”

Line 231 – Why is the background restricted to TAD? Motifs can be anywhere. Perhaps adding a minimal explanation in the main text instead of referring to methods would be a good clarification.

Response: We adjusted the text to:

“We compared all DC/IC/NC pairs that had ATAC-seq signal in both genomes relative **to control ATAC-peaks. To distinguish between orthologs and other enhancers of the same gene, the background consisted of non-orthologous ATAC-seq peaks within the same TAD (see Methods).**”

Re-assessment of heart-specific expression based on extended control experiments

We reevaluated the background staining stemming from the H11 locus during generation of additional enhancer reporters. Our setup was established at H11 locus based on a previous publication (Kvon et al. 2020, PMID: 32169219) and is designed to primarily profile limb enhancers. In prior experiments the empty landing pad and integrated limb enhancers we detected some background in the somites and otic vesicles, but no background in the heart (data available).

We now included a no-enhancer plasmid integration at the H11-landing pad as control and in addition to the background staining in somites and the otic vesicle, we detected faint, but specific signal in the outflow tract (OFT) of the E10.5 developing heart, but no additional expression in the ventricles or atria (**Fig. S7**). Thus, any signal detected in the OFT needs to be interpreted with caution.

We therefore reassessed the activity of all tested enhancers in the current version of the manuscript and present the controls and results in the revised results section (**I 302-330**) the **new Fig. 5a** and new supplemental **Fig. S7**.

Reviewer #3:

Remarks to the Author:

I congratulate the authors for the revision. All my main points have been addressed, and frankly I think the resulting manuscript is stronger than its original form (which was already very good). I have also looked at the github repository and that's clearly an improvement. Confirming that a 3rd party user tried this tool successfully is a valuable point.

I have one suggestion (see code section) of including an option for custom "fasta" genome files as input for the pipeline, as that could make this tool more useful (e.g. non-UCSC genomes, custom assemblies).

Then, I just wanted to clarify that when I asked "4 - Then, the authors should also do a motif analysis on the NC+ set of enhancers. Are these likely "novel" enhancers showing a different regulatory logic than the DC/IC? The TFBS conservation seems lower, but are the same motifs that dictate the general signal for the tissue also present there as the most enriched? Or the non-conserved enhancers are really the place to look for regulatory innovation?"

What I wanted to know, beyond what the overall conservation that figure 4e/f shows, is what kind of TFs bind to active non-conserved enhancers (NC+). NC+ are likely to be truly new (de novo) enhancers. So new enhancers could arise in two ways 1) gaining by chance a "core" heart TFBS and then everything forms from there. 2) just attracting new TFs, not particularly those of the "core" heart set, but some species-specific TF (at least species-specific in terms of expression domain). Focusing on conservation is interesting, and chicken vs mouse hearts have surely a lot in common. But they also have a lot not in common. So new traits might arise from new enhancers. And this is what I was asking, are NC+ gaining TFBS for TFs that are different across species? which ones? are they interesting or just a not very informative mix of zinc finger motifs? Lack of conservation suggest that, but are the divergent TFs interesting in any way? In sum, are NC+ where we should focus our efforts when trying to understand lineage-specific adaptations? This said, DC/IC enhancers surely also contribute to species variation (tinkering with old tools) but perhaps de novo enhancers have a different story to tell. I just make this clarification in case the authors find the answers to this question relevant to be included in this manuscript, but I am fine if this is left for later studies.

Remarks on code availability:

The github is well documented. My only suggestion here is that the tool is very focused to genomes that are available in UCSC, with named reference assemblies. However, in this day and age, single laboratory custom genomes are going to be more and more frequent. Also, UCSC is quite vertebrate biased, and given projects such as the Darwin Tree of Life, ERGA, etc, making this easily applicable to other lineages would be nice. Perhaps including the option (or some links to a guideline) to start with genome fasta files would be more broadly useable for the community.

RESPONSE:

We thank the reviewer for the positive feedback. While our setup is optimized for genome availability through UCSC, there is no limitation in providing self-assembled genomes.

In the output directory (defined by the -d flag), the pipeline will create a predefined folder structure to store relevant inputs like fasta files. The pipeline will try to look for genome fasta files from UCSC from the provided list of species (flag -s), but you can of course use your own custom genomes. In this case, store these fasta files under <output_directory_name>/fasta.

We updated the documentation on our github so this option is obvious to the user. You can find the updated version here at <https://github.com/mikstapes/IPP/> and on <https://github.com/tobiaszehnder/IPP>